# Characterization of Fibrodysplasia Ossificans Progessiva relevant Acvr1/Acvr2 Activin receptors in medaka (*Oryzias latipes*)

**Michael Trumpp**[1], **Wen Hui Tan**[2], **Wiktor Burdzinski**[1,3], **Yara Basler**[1], **Jerome Jatzlau**[1]*, **Petra Knaus**[1,3]*, **Christoph Winkler**[2]*

**1** Institute of Chemistry and Biochemistry, Freie Universität Berlin, Berlin, Germany, **2** Department of Biological Sciences and Centre for Bioimaging Sciences, National University of Singapore, Singapore, Singapore, **3** Berlin-Brandenburg School for Regenerative Therapies (BSRT), Berlin, Germany

* jerome.jatzlau@fu-berlin.de (JJ); knaus@zedat.fu-berlin.de (PK); dbswcw@nus.edu.sg (CW)

**Data Availability Statement:** All relevant data are within the paper and its Supporting Information files.

## Abstract

Activin and Bone Morphogenetic Protein (BMP) signaling plays crucial roles in vertebrate organ formation, including osteo- and angiogenesis, and tissue homeostasis, such as neuronal maintenance. Activin and BMP signaling needs to be precisely controlled by restricted expression of shared receptors, stoichiometric composition of receptor-complexes and presence of regulatory proteins. A R206H mutation in the human (hs) BMP type I receptor hsACVR1, on the other hand, leads to excessive phosphorylation of Sons of mothers against decapentaplegic (SMAD) 1/5/8. This in turn causes increased inflammation and heterotopic ossification in soft tissues of patients suffering from Fibrodysplasia Ossificans Progressiva (FOP). Several animal models have been established to understand the spontaneous and progressive nature of FOP, but often have inherent limitations. The Japanese medaka (*Oryzias latipes*, *ola*) has recently emerged as popular model for bone research. To assess whether medaka is suitable as a potential FOP animal model, we determined the expression of *Activin receptor type I* (*ACVR1*) orthologs *olaAcvr1* and *olaAcvr1l* with that of Activin type II receptors *olaAcvr2ab*, *olaAcvr2ba* and *olaAcvr2bb* in embryonic and adult medaka tissues by *in situ* hybridization. Further, we showed that Activin A binding properties are conserved in olaAcvr2, as are the mechanistic features in the GS-Box of both olaAcvr1 and olaAcvr1l. This consequently leads to FOP-typical elevated SMAD signaling when the medaka type I receptors carry the R206H equivalent FOP mutation. Together, this study therefore provides experimental groundwork needed to establish a unique medaka model to investigate mechanisms underlying FOP.

## Introduction

The growth factors Bone Morphogenetic Protein (BMP) and Activin are key components regulating organ formation, including osteo-, angio- and neurogenesis. They affect a wide range of diverse developmental processes, such as cell migration, proliferation and tissue

**Funding:** This work was funded by a NUS/BER Strategic Partnership grant (A-0004751-00-00) awarded from the Freie Universitaet Berlin and National University of Singapore to PK and CW, respectively. PK was also funded by German Research Foundation DFG (SFB 1444) and by IPSEN GmbH. CW is further funded by a grant from the Singapore Ministry of Education (MOE-T2EP30221-0008). The funders had no role in study design, data collection and analysis, decision to publish, or preparation of the manuscript.

**Competing interests:** The authors have declared that no competing interests exist.

homeostasis [1]. BMPs and Activins belong to the Transforming Growth Factor (TGF)-β ligand superfamily and signal through binding to a shared set of type I and type II serine/threonine transmembrane kinase receptors. Composition and localization of BMP receptor (BMPR) and Activin receptor (ACVR) complexes in different cellular and tissue environments are pivotal for the outcome and signaling specificity of BMP and Activin signaling [2]. After binding of a dimeric ligand to a hetero-tetrameric receptor complex, composed of two BMP type I and type II receptors, the constitutively active kinase of a type II receptor trans-phosphorylates the GS-box of a type I receptor and through this induces type I receptor kinase activity. SMAD specificity is defined by the type I receptors that activate either SMAD1/5/8 (e.g., ACVR1) or SMAD2/3 (e.g., ACVR1B) [3]. While BMPs signal through BMP type I receptors (e.g., ACVR1, ACVR1L) to activate SMAD1/5/8 responses [4, 5], Activins signal through Activin type I receptors (ACVR1B/C) to induce the SMAD2/3 branch [3]. Under physiological conditions, human Activin A signals through a complex comprising the high affinity type II receptor hsACVR2A/B and low affinity type I receptor hsACVR1B [6]. If Activin A, however, binds to a complex that contains hsACVR1, this results in a decoy complex that is unable to transmit SMAD1/5/8 signals [7–9]. As hsACVR1B competes with hsACVR1 for binding to common activin type II receptors, this provides a regulatory mechanism to control Activin A signal responses [10]. Only when activating mutations are present in the intracellular domain of hsACVR1, the decoy function of hsACVR1 is lost and Activin A induces SMAD1/5/8 signaling [11, 12]. For example, the most prominent hsACVR1 mutation in FOP, a substitution of arginine to histidine (R206H) in the glycine serine rich GS domain of the hsACVR1 kinase induces hyperactive SMAD1/5/8 signaling downstream of BMPs and Activin A [11, 13, 14]. Consequently, this imbalance causes the severe ultra-rare disease Fibrodysplasia Ossificans Progressiva (FOP), an autosomal dominant skeletal disorder characterized by widespread heterotopic bone formation in soft tissue [15].

The FOP-causing ACVR1 receptor was analyzed in mouse and zebrafish (*Danio rerio*, *dr*) to generate FOP models that mimic disease phenotype and progression [16, 17]. Single cell RNA sequencing of embryonic and adult mice and zebrafish revealed expression of *Acvr1*, *Bmpr2* and *Acvr2a/b* orthologs in chondrocytes, mesoderm, myocytes, endothelial cells, and osteocytes [18–21]. In both animal models, however, introduction of the ACVR1-R206H mutations resulted in perinatal or embryonic lethality. To avoid this, chimeric or conditional approaches were used, which in mice resulted in heterotopic ossification, hind limb digit malformations and joint fusions [11, 22]. In zebrafish, a heat-shock inducible *acvr1l_Q204D* model exhibited severe skeletal phenotypes similar to those reported in human FOP patients such as heterotopic ossification, spinal lordosis, vertebral fusions, and malformed pelvic fins [16]. However, due to the inducible nature of mutant ACVR1 overexpression in both models, the spontaneous and progressive features of FOP could only be partially mimicked. To fully recapitulate a human FOP phenotype, an ideal model requires (1) expression of an endogenous ACVR1 allele carrying the R206H mutation during development, (2) retaining the spontaneous nature of disease onset, and (3) exhibiting mild clinical features already at birth. Hence, to faithfully model progression of clinical FOP phenotypes from embryogenesis to adult stages, improved FOP animal models are needed that ideally carry a knock-in of FOP-relevant mutations directly in the endogenous *ACVR1* gene loci.

Similar to zebrafish, also medaka (*Oryzias latipes*, *ola*) produces abundant translucent embryos on a daily basis. These embryos develop rapidly, allow efficient genome manipulations and are accessible to high resolution live cell imaging and lineage tracing. We recently reported protocols for a highly efficient gene knock-in by homology directed repair (HDR) in medaka [23]. While zebrafish has one *drAcvr1l* ortholog, the medaka genome carries two *olaAcvr1* co-orthologs, *olaAcvr1* and *olaAcvr1l*. This offers a unique advantage for the

experimental use as FOP disease model. A second receptor copy is hypothesized to partially compensate for any effects caused by a modified receptor carrying the R206H-equivalent mutation. The presence of a second non-mutated receptor could thus potentially overcome embryonic lethality as seen in previously generated models, extend post-embryonic survival and thus better reflect the situation in human FOP patients.

In the present study, to understand which of the two medaka *olaAcvr1* co-orthologs can be used to generate a novel FOP knock-in mutant, we analyzed their embryonic as well as adult expression and compared this to the expression of BMP and Activin type II receptors. Specifically, we used RNA in-situ hybridization to determine co-expression of *olaAcvr1* and *olaAcvr1l* with that of Activin A high affinity type II receptors *olaAcvr2ab*, *olaAcvr2ba* and *olaAcvr2bb*. As controls, we analyzed *olaAlk1* encoding the ortholog of BMP type I receptor Activin receptor like 1 (ALK1/ACVRL1), which is the structurally closest relative of ACVR1, as well as the Activin A low affinity type II receptor *olaBmpr2a*. This identified several tissues where individual receptors were co-expressed, suggesting possible complex formation, while other receptors showed restricted expression, e.g., in distinct layers of the retina. Furthermore, we determined the degree of hsActivin A ligand binding to medaka olaAcvr2ab, olaAcvr2ba and olaAcvr2bb. We also introduced a GS-BOX destabilizing mutation equivalent to R206H into olaAcvr1 and olaAcvr1l. In both cases, this resulted in elevated SMAD1/5/8 signaling, which is a typical hallmark of the FOP pathology.

## Results

### Mapping of *olaAcvr1*/*olaAcvr2* co-expression domains in the medaka central nervous system

To localize regions of *olaAcvr1*/*olaAcvr2* co-expression in developing and adult medaka, RNA *in situ* hybridization was performed for *olaAcvr1*, *olaAcvr1l*, *olaAcvr2ab*, *olaAcvr2ba* and *olaAcvr2bb* in whole embryos at 5 days post fertilization (dpf) and on transverse sections of adult medaka heads at 2 months post fertilization (mpf). Expression of *olaAlk1* and *olaBmpr2a* were used for comparison. The specificity of riboprobes was validated by comparing 5 dpf embryos stained with anti-sense and sense riboprobes (**S1 Fig**).

In 5 dpf embryos, we observed co-expression of type I (*olaAcvr1* and *olaAcvr1l*) and type II (*olaAcvr2ab*, *olaAcvr2ba*, *olaAcvr2bb*) receptors at the mid-hindbrain boundary (mhb), in the cerebellum, branchial arches and the midline of the developing fore- and midbrain where forming ventricles are located (**Fig 1A–1E'**). Ventral views revealed subtle differences in the expression domains of co-orthologous receptors. Within type I receptors, *olaAcvr1* formed a distinct pattern along the midline of fore-, mid- and hindbrain (**Fig 1A'**). In contrast, *olaAcvr1l* showed a broader expression pattern in similar regions at the mhb and cerebellum, and also elevated levels in the branchial arches (**Fig 1B'**). Similarly, type II receptor orthologs also showed differential expression patterns. While *olaAvr2ab* and *olaAcvr2bb* showed a broader and less confined pattern throughout the developing brain, with elevated levels in cerebellum and mhb, *olaAcvr2ba* showed strong but more confined expression in the same regions (**Fig 1C'–1E'**). The type II receptor paralogs *olaAcvr2ab* and *olaAcvr2ba* were also highly expressed in the branchial arches (**Fig 1C'** and **1D'**). Medaka *olaAlk1* was expressed in blood vessels in close vicinity to skeletal structures such as the ceratohyal, branchial arches, basibranchial, basihyal and palatoquadrate (**S2A and S2A' Fig**).

RNA *in situ* hybridization of cryo-sectioned brains of medaka at 2 mpf revealed expression of all tested type I and type II receptors in the proliferation zones surrounding the ventricles of the telencephalon (**Fig 1F–1J**). While the type I receptor expression domains generally corresponded well to those identified at the 5 dpf embryo stage, *olaAcvr1* showed a more confined

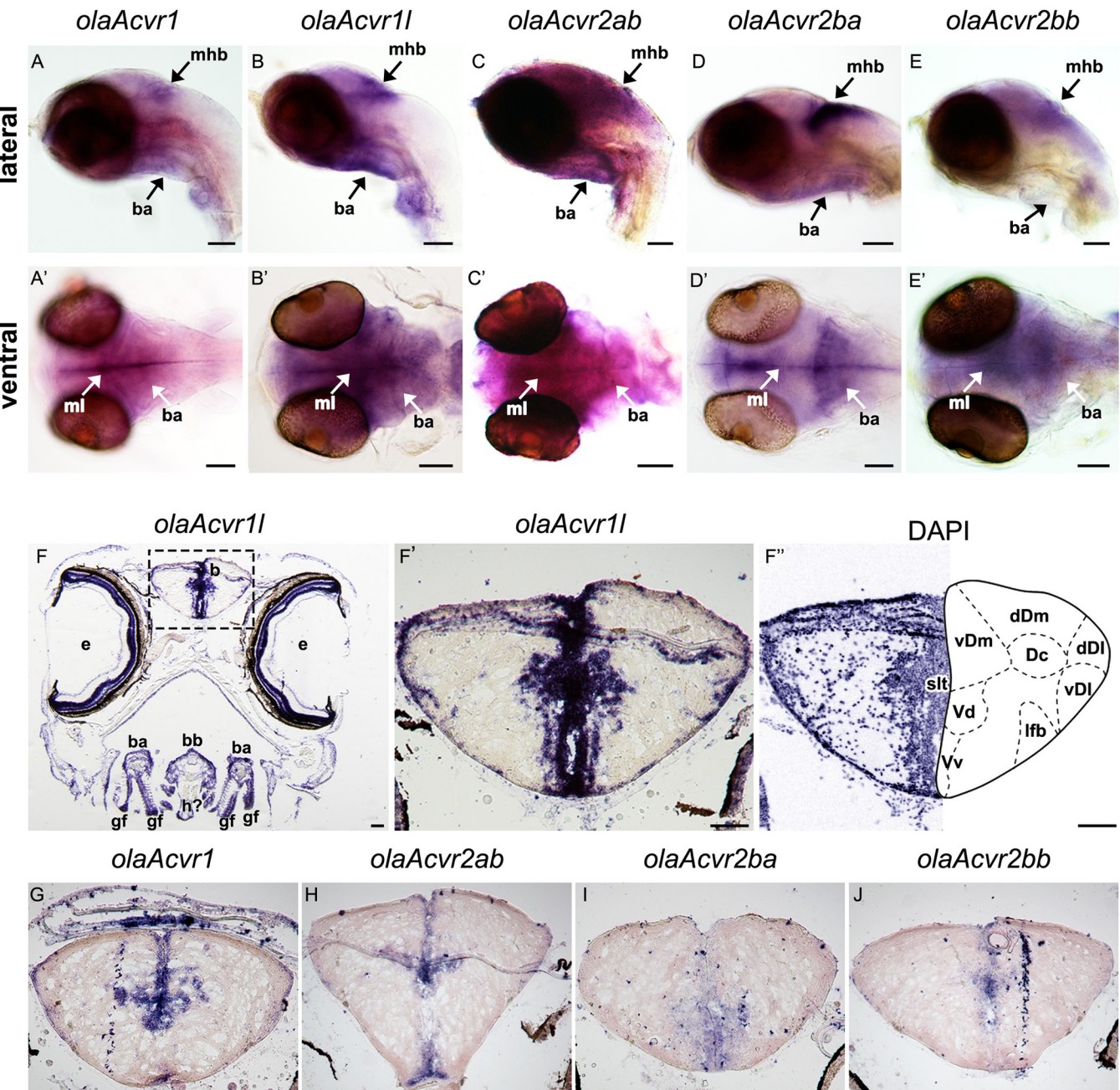

**Fig 1. Expression of *olaAcvr1*, *olaAcvr1l*, *olaAcvr2ab*, *olaAcvr2ba*, *olaAcvr2bb* in medaka embryos and the adult telencephalon.** (A-E') Comparison of *olaAcvr1*, *olaAcvr1l*, *olaAcvr2ab*, *olaAcvr2ba* and *olaAcvr2bb* expression by whole mount RNA *in situ* hybridization in 5 dpf medaka embryos. (A-E) Lateral views of medaka embryonic heads. (A'-E') Ventral views of medaka embryonic heads. (F-J) Comparison of *olaAcvr1*, *olaAcvr1l*, *olaAcvr2ab*, *olaAcvr2ba* and *olaAcvr2bb* expression in 2 mpf medaka brains after RNA *in situ* hybridization on cryo-sections. (F) Overview of *olaAcvr1l* stained head section. (F') Zoom in on *olaAcvr1l* stained brain tissue. (F") Respective nuclear staining (DAPI; pseudo-colored) of (F') and schematic diagram of telencephalon regions based on [56]; b–brain, ba–branchial arches, bb–basibranchial, Dc–area dorsalis telencephali pars centralis, dDl–dorsal region of area dorsalis telencephali pars lateralis (Dl), dDm–dorsal region of Dm, e–eye, gf–gill filaments, mhb–mid-hindbrain, ml–midline, lfb—lateral forebrain bundle (fasciculus lateralis telencephali), Slt–sulcus limitans telencelphali, vDl—ventral region of Dl, vDm–ventral region of area dorsalis telencephali pars medialis (Dm), Vd–area ventralis telencephalic pars dorsalis, Vv–area ventralis telencephalic pars ventralis. Scale bars = 100 µm.

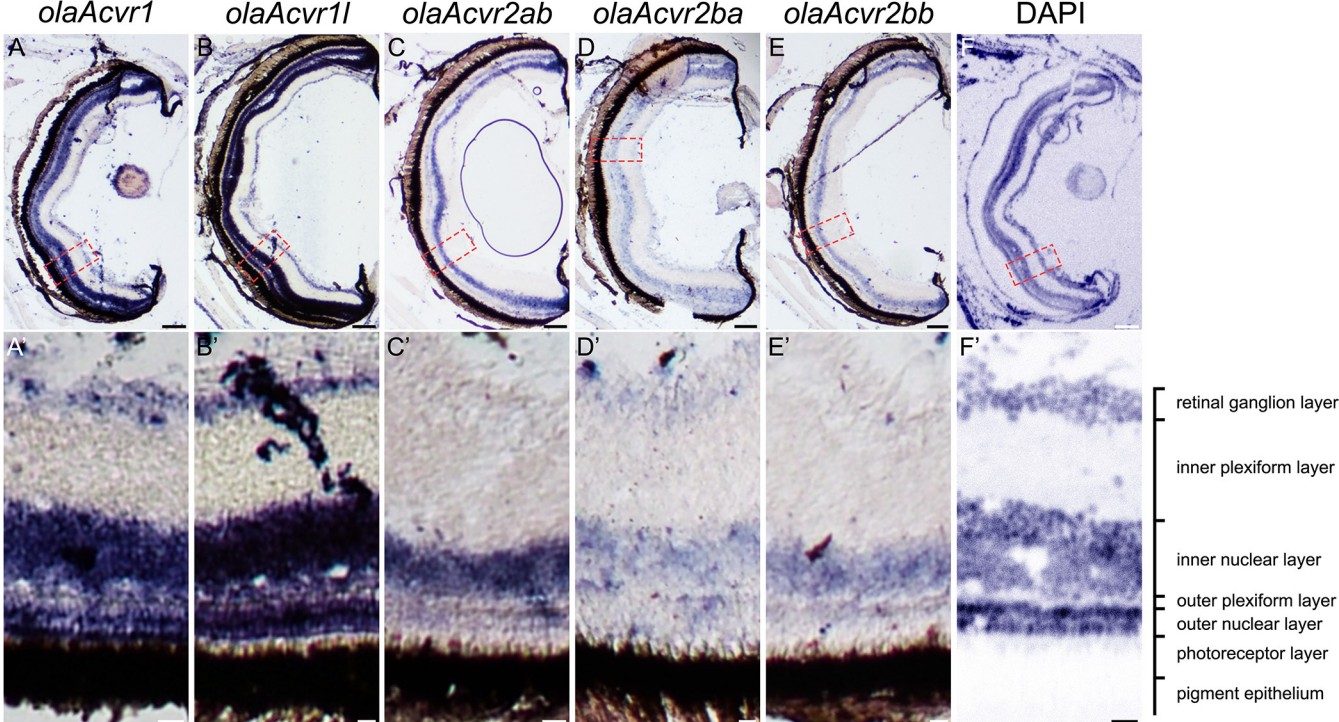

**Fig 2. Expression of *olaAcvr1*, *olaAcvr1l*, *olaAcvr2ab*, *olaAcvr2ba* and *olaAcvr2bb* in the adult medaka retina.** Comparison of *olaAcvr1*, *olaAcvr1l*, *olaAcvr2ab*, *olaAcvr2ba* and *olaAcvr2bb* expression patterns in eyes of 2 mpf medaka by RNA *in situ* hybridization on cryosections. (A-E) Overview of respective receptor mRNA staining on eye sections. (A'-E') Zoom in on respectively stained retinal tissue. (F, F') Nuclear staining (DAPI; pseudo-colored) of eye tissue with description of eye layers. Scale bars (A-F) = 100 μm, (A'-F') = 10 μm.

expression pattern when compared to its broadly expressed co-ortholog *olaAcvr1l* (**Fig 1F' and 1G**). Expression of the structurally related type I receptor *olaAlk1* was also observed around ventricles (**S2B and S2C Fig**). The type II receptor paralogs *olaAcvr2ab*, *olaAcvr2ba* and *olaAcvr2bb* were expressed in a more restricted manner in the same regions (**Fig 1H–1J**).

Besides expression in subventricular neuron progenitor zones of the forebrain, all investigated receptors were also found to be expressed in the retina at 2 mpf (**Fig 2**). *olaAcvr1* and *olaAcvr1l* mRNA staining revealed expression in the retinal ganglion cell layer, as well as throughout the entire inner and outer nuclear layers, the latter containing cone and rod photoreceptors (**Fig 2A and 2B'**). In contrast, the type II orthologs *olaAcvr2ab*, *olaAcvr2ba* and *olaAcvr2bb* showed less pronounced expression in the ganglion cell and outer nuclear layer, while there was considerable *olaAcvr2ab* expression in the inner nuclear layer containing bipolar cells (**Fig 2C–2E'**). On the other hand, *olaAlk1* and *olaBmpr2a* were both expressed in the inner and outer nuclear layers as well as in the retinal ganglion cell layer (**S3 Fig**).

## Mapping of *olaAcvr1*/*olaAcvr2* co-expression in gills

Expression of BMP receptors in medaka is not limited to neuronal tissue. In embryonic gills at 5 dpf, *olaAlk1*, *olaAcvr1*, *olaAcvr1l*, *olaAcvr2ab* and *olaAcvr2ba* showed expression in branchial arches (**Fig 1A–1E', S1A and S1A' Fig**). In gills of adult fish (2 mpf), *olaAlk1*, *olaAcvr1*, *olaAcvr1l*, *olaAcvr2ab*, *olaAcvr2ba*, *olaAcvr2bb* and *olaBmpr2a* showed expression in branchial arches, basibranchials, basihyals and gill filaments (**Fig 3A–3E, S4A–S4H Fig**). Whereas *olaAcvr1* expression was most prominently found in endothelial pillar cells of the efferent arteries in gill filaments, its co-ortholog *olaAcvr1l* was broadly expressed within both the

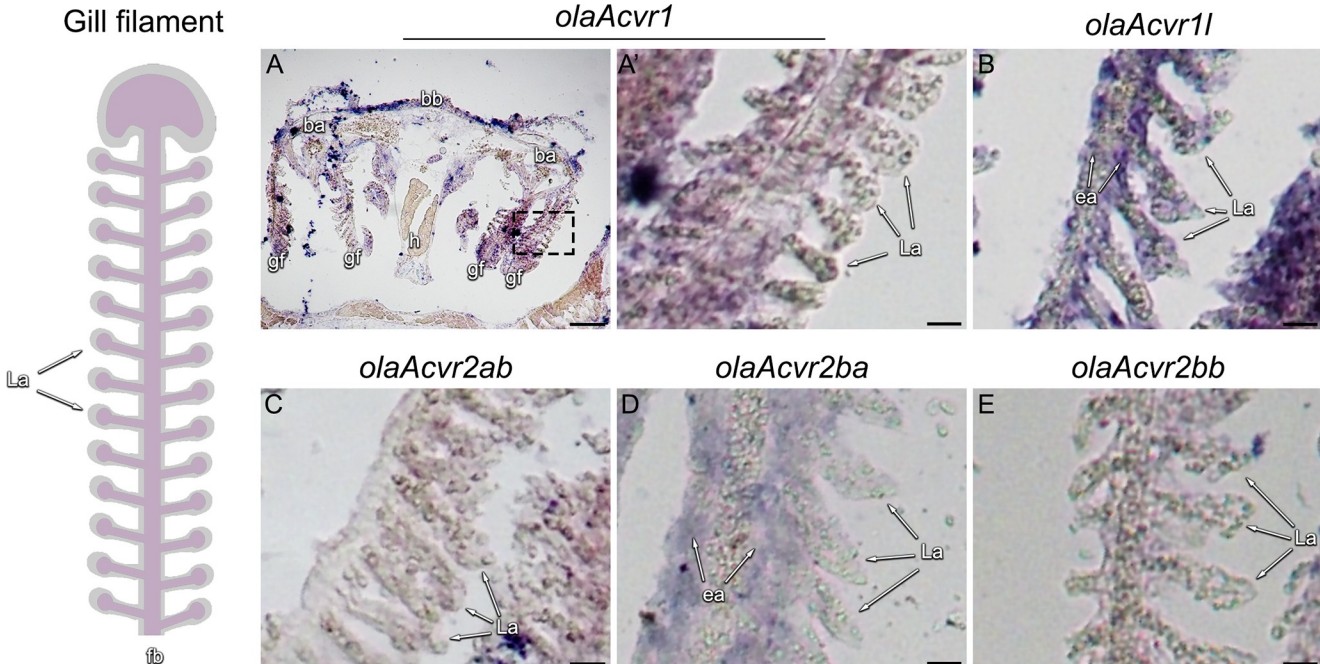

**Fig 3. Expression patterns of *olaAcvr1*, *olaAcvr1l*, *olaAcvr2ab*, *olaAcvr2ba* and *olaAcvr2bb* in adult medaka gills.** Sketch of an intact medaka gill filament (left). (A) Overview of *olaAcvr1* expression patterns in gill tissue. (A'-E) Zoom-in of *olaAcvr1*, *olaAcvr1l*, *olaAcvr2ab*, *olaAcvr2ba* and *olaAcvr2bb* expression patterns in 2 mpf medaka gill filaments by RNA *in situ* hybridization on cryosections. ba–branchial arch, bb–basibranchial, ea–efferent artery, fb–filament base, gf–gill filaments, h–heart, L–lamellae. Scale bar (A) = 100 μm, (B-E) = 15 μm.

lamellae and efferent arteries of the gill filaments (**Fig 3A and 3B**). While *olaAcvr2ab* and *olaAcvr2bb* showed no obvious staining in gill filaments, *olaAcvr2ba* was predominantly expressed in the efferent arteries of the gill filaments but less pronounced in the lamellae (**Fig 3C–3E**). In contrast, *olaAlk1* and *olaBmpr2a* were both expressed in lamellae and efferent arteries of the gill filaments, with *olaBmpr2a* showing a much more confined pattern (**S4 Fig**).

## Mapping of *olaAcvr1/olaAcvr2* co-expression in the medaka trunk

Whole mount RNA *in situ* hybridization of the trunk and pectoral fins of 5 dpf medaka embryos revealed distinct expression patterns for the type I *olaAcvr1* and *olaAcvr1l*, and type II *olaAcvr2ab*, *olaAcvr2ba* and *olaAcvr2bb* co-orthologs (**Fig 4A–4J**). While no obvious expression was detected for *olaAcvr1l* in the trunk, weak *olaAcvr1* staining was found in both dorsal and ventral regions of the trunk. (**Fig 4A and 4B**). In contrast, *olaAcvr2ab* was found strongly expressed both dorsally and ventrally (**Fig 4C**). Importantly, *olaAcvr2ba* and *olaAcvr2bb* were expressed in a mutually exclusive pattern in either the dorsal or ventral part of the trunk, respectively (**Fig 4D and 4E**). Similar to *olaAcvr2ba*, *olaAlk1* expression was found in the dorsal part, while *olaBmpr2a* was expressed in the ventral part of the posterior body (**S5A and S5B Fig**).

In addition, type I receptors were expressed similarly in the pectoral fin bud (**Fig 4F and 4G**), while type II receptors were expressed with slightly different patterns (**Fig 4H–4J**). *olaAcvr2ab* was broadly expressed whereas *olaAcvr2bb* was expressed in a more confined manner in the anterior fin region (**Fig 4H and 4J**). In contrast, *olaAcvr2ba* showed weak if any expression (**Fig 4I**). Staining of all analyzed receptor mRNAs, including olaAlk1 and olaBmpr2a, showed signal in the fin mesenchyme but was excluded from the apical ectodermal ridge (aer; **Fig 4F–4J, S5C and S5D Fig**). In adult medaka at 2 mpf, *in situ* hybridization on trunk

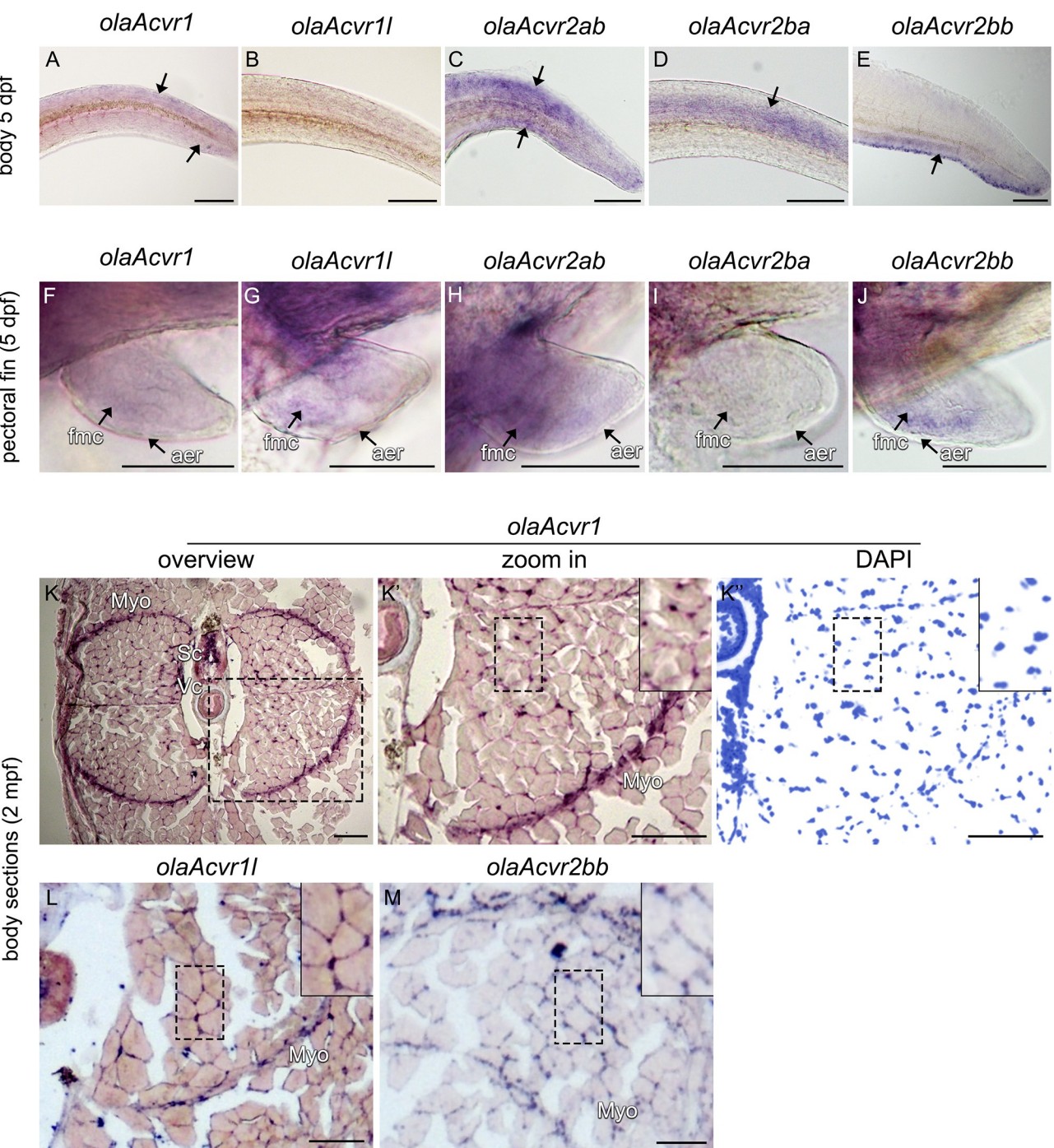

**Fig 4. Expression patterns of *olaAcvr1*, *olaAcvr1l*, *olaAcvr2ab*, *olaAcvr2ba* and *olaAcvr2bb* in the trunk of medaka embryos and adult cryo-sections.** (A-J) Comparison of *olaAcvr1*, *olaAcvr1l*, *olaAcvr2ab*, *olaAcvr2ba* and *olaAcvr2bb* expression by whole mount RNA *in situ* hybridization in 5 dpf medaka embryos. (A-E) Comparison of *olaAcvr1*, *olaAcvr1l*, *olaAcvr2ab*, *olaAcvr2ba* and *olaAcvr2bb* expression patterns in trunk and tail. (F-J) Comparison of *olaAcvr1*, *olaAcvr1l*, *olaAcvr2ab*, *olaAcvr2ba* and *olaAcvr2bb* expression in the pectoral fins of 5 dpf embryos. (K-M) *olaAcvr1*, *olaAcvr1l* and *olaAcvr2bb* expression after RNA *in situ* hybridization on cryosections of 2 mpf medaka trunks. (K) Overview of *olaAcvr1* expression in 2 mpf trunk sections. (K'-K") zoom in on *olaAcvr1* stained tissue (region indicated by box in K), and respective nuclear staining (DAPI; pseudo-colored). (L, M) Expression of olaAcvr1l and olaAcvr2bb expression in similar regions. Areas marked by dashed boxes are shown in higher magnification in the top right corners, respectively. aer–apical ectodermal ridge, fmc–fin mesenchyme, Myo–myosepta, Sc–spinal cord, Vc–vertebral column. Scale bars = 100 μm.

**Table 1. Summary of *olaAcvr1, olaAcvr1l, olaAcvr2ab, olaAcvr2ba, olaAcvr2bb, olaAlk1 and olaBmpr2a* expression in embryonic (5 dpf) and adult (2 mpf) medaka tissues (+ = expressed, n.d. = not detectable).**

| | *Acvr1* | *Acvr1l* | *Acvr2ab* | *Acvr2ba* | *Acvr2bb* | *Alk1* | *Bmpr2a* |
|---|---|---|---|---|---|---|---|
| **Head (5 dpf)** | | | | | | | |
| Cerebellum | + | + | + | + | + | n.d. | + |
| Mid-hindbrain boundary | + | + | + | + | + | + | + |
| Branchial arches | + | + | + | + | n.d. | + | n.d. |
| **Trunk (5 dpf)** | | | | | | | |
| Ventral | + | n.d. | + | n.d. | + | n.d. | + |
| Dorsal | + | n.d. | + | + | n.d. | + | n.d. |
| **Pectoral fin (5 dpf)** | | | | | | | |
| Apical ectodermal ridge | n.d. | n.d. | n.d. | n.d. | n.d. | n.d. | n.d. |
| Fin mesenchyme | + | + | + | + | + | + | + |
| **Telencephalon (2 mpf)** | | | | | | | |
| Proliferation zones | + | + | + | + | + | + | + |
| **Retina (2 mpf)** | | | | | | | |
| Retinal ganglion layer | + | + | n.d. | + | n.d. | + | + |
| Inner plexiform layer | n.d. | n.d. | n.d. | n.d. | n.d. | n.d. | n.d. |
| Inner nuclear layer | + | + | + | + | + | + | + |
| Outer plexiform layer | + | + | + | + | + | + | + |
| Outer nuclear layer | + | + | + | + | + | + | + |
| Photoreceptor layer | + | + | + | + | + | + | + |
| Retinal pigment epithelium | n.d. | n.d. | n.d. | n.d. | n.d. | n.d. | n.d. |
| **Gills (2 mpf)** | | | | | | | |
| Basibranchial | + | + | + | + | + | + | + |
| Branchial arches | + | + | + | + | + | + | + |
| Gill filaments | + | + | n.d. | + | n.d. | + | + |
| Efferent arteries | + | + | n.d. | + | n.d. | + | + |
| Lamellae | n.d. | + | n.d. | n.d. | n.d. | + | + |
| **Trunk (2 mpf)** | | | | | | | |
| Myoseptal cells | + | + | n.d. | n.d. | + | + | n.d. |
| Muscle interstitial cells | + | + | n.d. | n.d. | + | + | + |
| Vertebral bodies | n.d. | n.d. | n.d. | n.d. | n.d. | n.d. | + |

cryosections revealed strong expression of *olaAcvr1*, *olaAcvr1l*, *olaAcvr2bb* and *olaAlk1* in myoseptal cells and muscle interstitial cells while *olaAcvr2ab*, *olaAcvr2ba* and *olaBmpr2a* showed no obvious signal in this region (**Fig 4K–4M**, **S5E–S5I Fig**). Together, these results suggest that *olaAcvr1* and *olaAcvr1l*, the medaka orthologs of FOP-causing *hsACVR1*, as well as *olaAcvr2ab*, *olaAcvr2ba* and *olaAcvr2bb*, the orthologs of *hsACVR2A/B* encoding the corresponding interacting type II receptor, are mostly co-expressed in embryonic and adult tissues (summarized in **Table 1**). Most notably, they are expressed in soft tissues that get predominantly affected by FOP in humans.

## Conserved Activin type II receptor binding capacities

To investigate whether type II receptor domains that are implicated in human FOP pathology are conserved in medaka, we analyzed ligand binding domains (LBDs) by sequence comparison and performed a ligand surface binding assay (LSBA) [24] using fluorescently labelled hsActivin A-Cy5 (**Fig 5**, **S6 Fig**). Sequence comparison of the hsACVR2A LBD with that of olaAcvr2ab highlighted subtle differences in two hydrophobic amino acids (Phe61 hsACVR2A

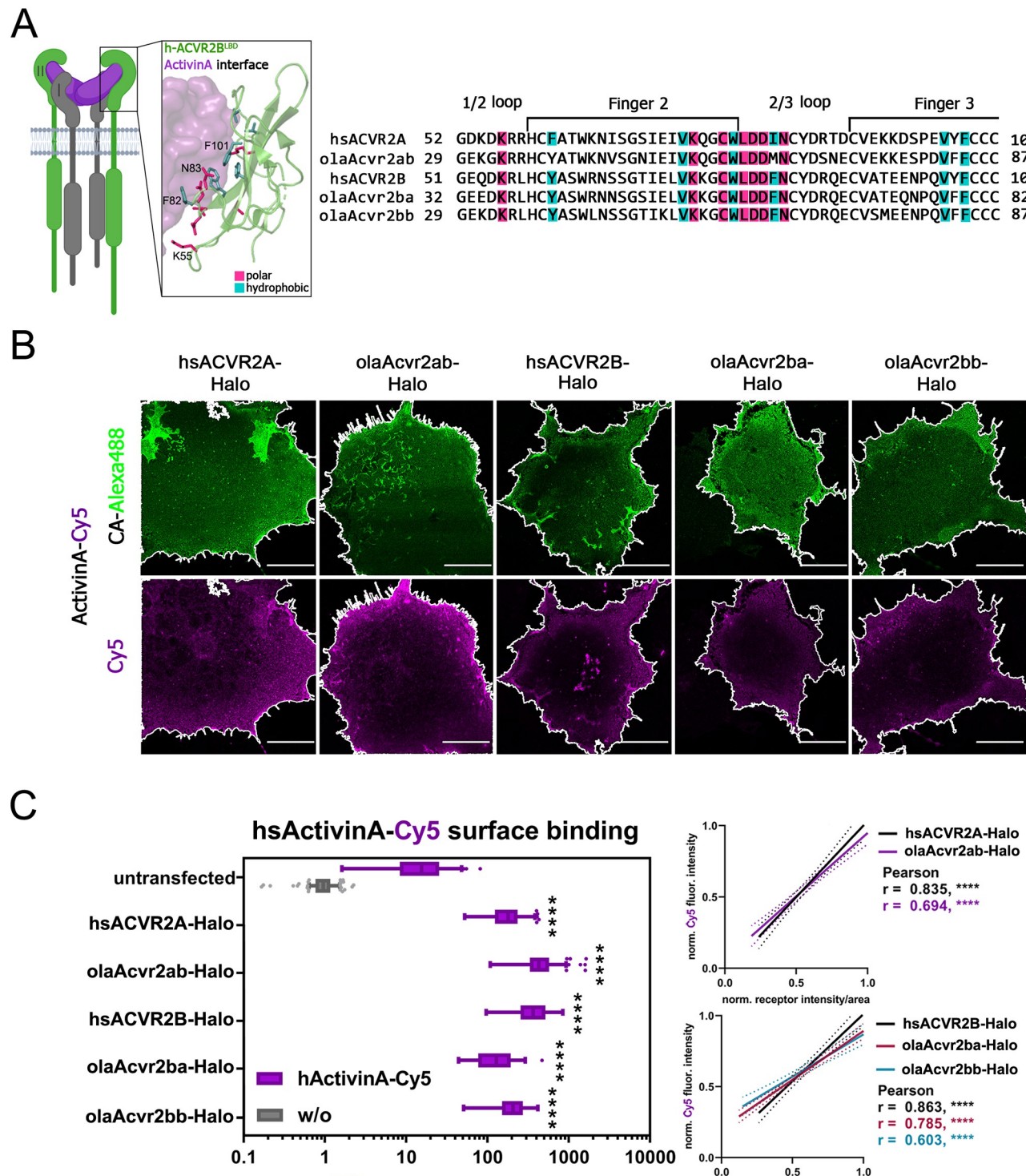

**Fig 5. Comparison of hsActivinA binding capabilities to human and medaka type II BMPRs.** (A) hsACVR2A/B ligand binding domain sequence alignment with respective medaka receptors; polar receptor:ligand interaction sites are shown in light magenta and hydrophobic interaction sites in teal (PDB 1NYU). (B-C) Transiently transfected COS-7 cells expressing Halo-tagged hsACVR2A, hsACVR2B, olaAcvr2ab, olaAcvr2ba and olaAcvr2bb receptors were simultaneously incubated with Halo-tag substrate CA-Alexa488 (green) and hsActivin A-Cy5 (magenta) at 24 hours post transfection (n = 3 independent experiments). (B) Representative confocal microscopy images of COS-7 cells transiently expressing respective receptors. Scale bar = 20 μm. (C left) hsActivin A-Cy5 surface binding represented as relative fluorescence intensity per area to untransfected unstimulated control. Data is shown as fold induction ± standard deviation. Negative controls are shown in S6 Fig. Significance was calculated using two-way ANOVA and Dunett

post hoc test comparing Cy5 intensity relative to untransfected cells. ****p < 0.001 ≡ significance as indicated (n = 3). (C right) Linear regression and correlation analysis of ligand:receptor binding based on Cy5-fluorescence intensity and normalized receptor fluorescence (CA-Alexa488) per area (n = 3). Correlation was analyzed using the Pearson Test (p < 0.0001 ≡ ****).

/ Tyr61 olaAcvr2ab, Ile83 hsACVR2A / Met83 olaAcvr2ab). These residues are known to form direct hydrophobic bonds between the human ligand and human receptors [25]. Comparison of hsACVR2B with olaAcvr2ba and olaAcvr2bb revealed that all polar and hydrophobic Activin A contact sites were identical. Similar to hsACVR2A, hsACVR2B and olaAcvr2b are highly conserved (Fig 5A). Further sequence comparison of hsActivin A and the medaka orthologs showed a similar conserved binding interface with a slight difference in two amino acids (Phe 327 hsActivin A / Tyr358 olaActivin Aa / Tyr 299 olaActivin Ab and Phe 407 hsActivin A / Tyr 438 olaActivin Aa/ Tyr 379 olaActivin Ab) (S6A Fig). Together, this suggests that Activin A binding to medaka olaAcvr2ab, olaAcvr2ba and olaAcvr2bb is likely to be similar as in the human counterparts. To test this hypothesis, we performed LSBA [24] using COS7-cells that transiently expressed Halo-tagged hsACVR2A, hsACVR2B, and olaAcvr2ab, olaAcvr2ba and olaAcvr2bb. As expected from previous studies, strong hsActivin A-Cy5 binding was observed for cells expressing hsACVR2A-Halo and hsACVR2B-Halo but not the non-binding receptor hsTGFBR2-Halo (Fig 5B, S6B Fig). Transient expression of medaka type II receptors also led to increased human hsActivin A-Cy5 ligand binding, similar to the respective human orthologs (Fig 5C). Furthermore, hsActivin A-Cy5 and Halo-tagged receptor intensities (AF488) positively correlated for all human and medaka type II receptors (Fig 5C). These results indicate that the LBDs of all tested type II receptors are highly conserved between human and medaka, and that human Activin A effectively binds to the medaka type II receptors.

## Elevated SMAD signaling induced by medaka olaAcvr1 and olaAcvr1l receptors carrying a R206H-equivalent mutation

To further investigate whether any of the two medaka olaAcvr1 receptors can be used to mimic the FOP signaling scenario by introduction of a R206H mutation, we next analyzed the cytosolic domain of the receptor, particularly the glycine/serine rich region (GS-box), by sequence comparison and Western blot analysis using pSMAD antibodies as well as reporter gene assays (Fig 6A–6E). To activate the kinase function of a type I receptor, the GS-box needs to be phosphorylated by a type II receptor. The R206H mutation lies within this GS-box of hsACVR1 and leads to hyper-activation of the receptor in FOP [26]. A sequence alignment of the GS-box in hsACVR1 with medaka olaAcvr1 and olaAcvr1l revealed high conservation between both species. Specifically, the olaAcvr1 GS-box sequence was found to be identical to human, and olaAcvr1l had only one amino acid change (His219 olaAcvr1l / Gln219) (Fig 6A). Furthermore, a sequence comparison of the entire kinase domain showed 90.6% conservation of olaAcvr1 and 82.8% conservation of olaAcvr1l to human hsACVR1 (S7A Fig). In contrast, the sequence comparison of the type I LBD revealed a significantly lower conservation with 61.8% in olaAcvr1 and 40.0% in olaAcvr1l to hsACVR1 (S7B Fig). Interestingly, the amino acid sequence comparison also revealed that the LBD of olaAcvr1 was significantly longer than that of its paralog olaAcvr1l. In comparison to human *hsACVR1*, both medaka *olaAcvr1* and *olaAcvr1l* genes encode at least two different transcripts, which are both characterized by an elongated N-terminus. While olaAcvr1 is predicted to contain three ligand binding domains, olaAcvr1l has an unstructured N-terminus (schematically illustrated in Fig 6B). To characterize both olaAcvr1 variants, we cloned full-length versions of both receptor cDNAs from medaka (18 dpf) and generated mutated FOP versions by exchanging the conserved R206-equivalent amino acid R379 of olaAcvr1 and R239 of olaAcvr1l to Histidine. Thus, the

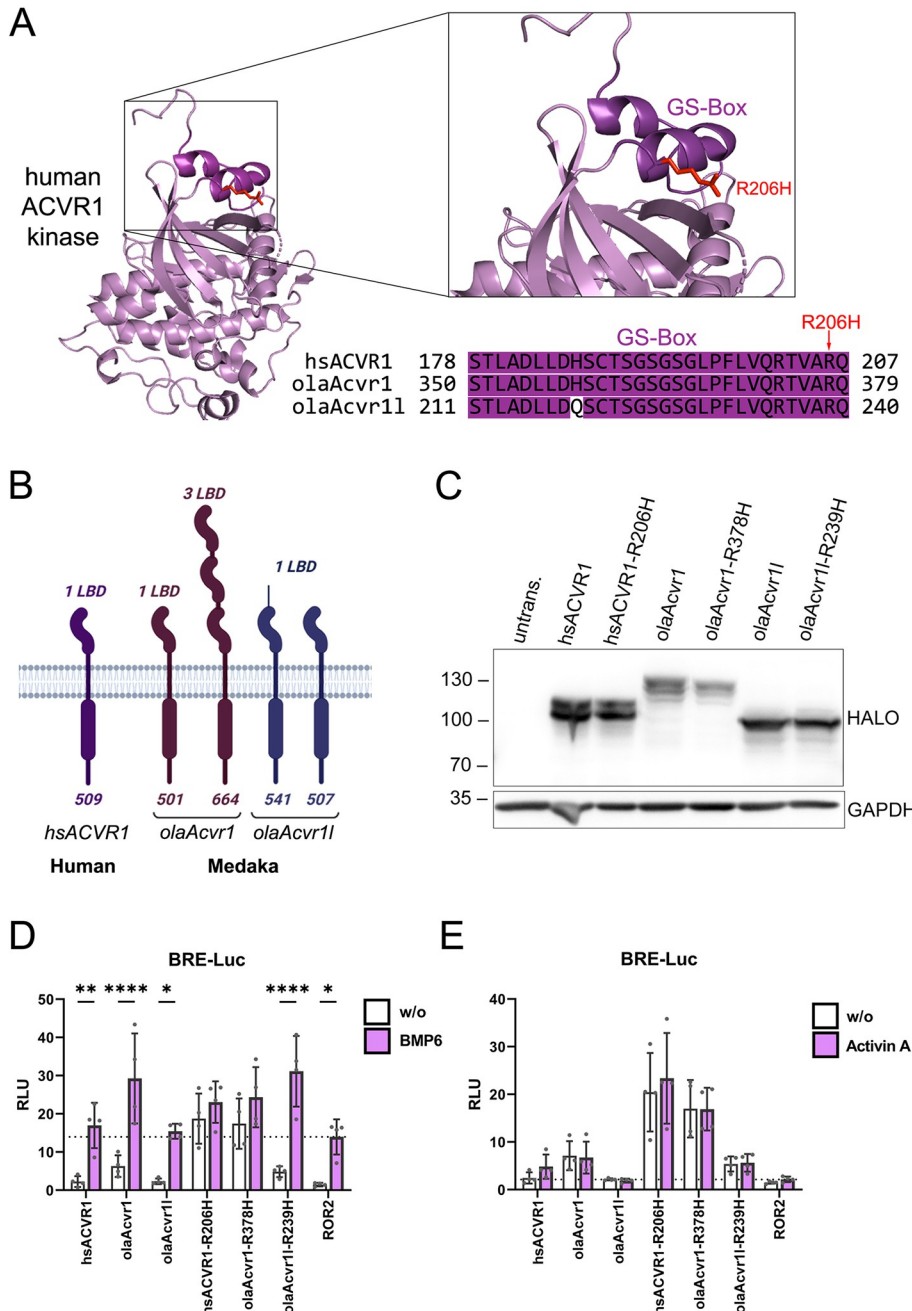

**Fig 6. Comparison of human and medaka ACVR1 receptors.** (A) Alignment of the hsACVR1 GS-Box with that of respective medaka receptors; similarities are highlighted in magenta (PDB 6I1S). (B) Schematic drawing of hsACVR1 and olaAcvr1 and olaAcvr1l. (C) Western blot analysis of Halo tagged hsACVR1, hsACVR1-R206H, olaAcvr1, olaAcvr1-R378H, olaAcvr1l and olaAcvr1l-R239H, transiently expressed in HEK293T cells with GAPDH as loading reference. All receptors showed the expected weight shift according to the Halo-tag size (~32 kDa). (D, E) Luciferase activity one day after transfection with a SMAD1/5/8-sensitive BRE2-luciferase reporter and hRluc/TK together with Halo tagged hsACVR1, hsACVR1-R206H, olaAcvr1, olaAcvr1-R378H, olaAcvr1l, olaAcvr1l-R239H and ROR2 as control for endogenous signaling. HEK293T cells were starved for 3 h and stimulated overnight with BMP6 (2 nM) or Activin A (5 nM). Relative Luminescence Units (RLU) are expressed as mean fold induction ± standard deviation (n = 4 independent experiments). Statistical significance relative to no-stimulation control (w/o) was calculated using two-way ANOVA and Šidák multiple comparison test post-hoc test; *P< 0.05, **P< 0.01, ****P<0.0001.

introduced olaAcvr1-R379H and olaAcvr1l-R239H mutations exactly recapitulated human hsACVR1-R206H. We verified expression of all constructs in transiently transfected HEK293T cells by Western blot (**Fig 6C**). While Halo-tagged hsACVR1, olaAcvr1l and their respective R206H mutants showed molecular weights of 100–110 kDa, olaAcvr1 showed a higher molecular weight of 130 kDa reflecting the presence of three LBDs. In comparison to hsACVR1 and hsACVR1-R206H, all expressed medaka receptors were then tested for their capability to transmit BMP6 and Activin A signals using a BRE luciferase reporter system (**Fig 6D and 6E**). Replacement of Arginine to Histidine in the GS-box of both olaAcvr1 and olaAcvr1l resulted in elevated BRE-reporter activity similar to that seen for human hsACVR1. While unstimulated hsACVR1-R206H and olaAcvr1-R378H induced similar luciferase activity, olaAcvr1l-R239H showed lower levels. BMP6 stimulation of hsACVR1 and olaAcvr1l showed no elevated luciferase intensity compared to a control plasmid (ROR2-expressing), while olaAcvr1 and olaAcvr1l-R239H showed an increase in reporter activity (**Fig 6D**). For all receptors, stimulation with Activin A did not lead to increased luciferase signal intensity (**Fig 6E**). Concomitant expression of a type II receptor did not further increase hsActivin A induced BRE-LUC activity (**S7C Fig**).

## Discussion

The Japanese medaka (*Oryzias latipes*) has recently emerged as a popular model for skeletal research [27–29]. Pathways for bone development, as well as molecular profiles of bone regulating cells are highly conserved between mammals and teleost fish [30]. Several studies from our and other labs highlighted the suitability of medaka as a model organism to study bone formation, remodeling, regeneration and bone diseases such as osteoporosis [31–33]. To generate a novel FOP disease animal model, we assessed and compared the embryonic and adult expression of *olaAcvr1*, *olaAcvr1l*, *olaAcvr2ab*, *olaAcvr2ba*, *olaAcvr2bb* as well as *olaAlk1* and *olaBmpr2a* in several tissues of medaka by *in situ* hybridization. Further, we investigated via sequence comparison, Western blot analysis, ligand surface binding assays and reporter gene assays the conserved signaling capacities of the medaka and human receptors for both wild-type and FOP-related receptor mutations.

### Expression of BMPRs in neuronal tissues

In the developing as well as adult brain, neural stem cells (NSCs) proliferate in niches of the subventricular zones, where they subsequently differentiate into neurons and other progenitor cells. Over the last two decades, important roles have emerged for BMP signaling in NSC proliferation and differentiation. In mouse and chick, for example, it was reported that BMP signaling regulates stem cell specification in the developing cerebellum [34]. Further, in adult zebrafish it was shown that BMP signaling is detrimental for the maintenance of NSC pools in the subventricular zones of the telencephalon, by promoting the downstream transcription factor "inhibitor of differentiation 1" *(id1)* via SMAD 1/5 signaling, and thereby promoting the number of quiescent NSCs. In that study, BMPs were mainly found to be expressed in neurons around the NSCs of the subventricular proliferation zones [35]. It was proposed that neurons are BMP-producing and that NSCs are BMP-sensing cells. Consistent with this idea, it was shown that Activin A drives differentiation of developing rat cerebrocortical neural progenitor cells (NPCs) towards a neuronal fate [36]. Furthermore, previous studies highlighted the role of Activin A as a potent neurotrophic factor in regulating differentiation and identity of telencephalic neural precursors derived from human and mouse ESCs via inhibiting Sonic hedgehog (Shh) signaling. The underlying mechanisms remain unknown but it was proposed that Smad 2 interacts with GLi3R, a Shh inhibiting transcription factor [37]. Additionally,

Activin A enhances the response to retinoic acid signaling and thereby exerts its neurogenic effects [38]. Together, this shows that BMP signaling controls neurogenesis, albeit by different mechanisms.

In the present study, *in situ* hybridization revealed expression of *olaAcvr1*, *olaAcvr1l*, *olaAcvr2ab*, *olaAcvr2ba*, *olaAcvr2bb* and *olaBmpr2a* in neurogenic zones, as well as the mid-hindbrain and cerebellum of 5 dpf embryos. Receptor expression in these areas is consistent with previous findings that support important roles of BMP signaling for brain development by regulating neural progenitor differentiation. It can be speculated that the respective olaAcvr1 receptors form complexes with type II receptor variants of olaAcvr2 or Bmpr2, which under Bmp stimulation induce Smad1/5/8 signaling to drive neuronal differentiation. Also, an Activin signaling receptor complex with additional players like type I receptors Alk4 or Alk7 could promote Smad2/3 signaling and by this balance neuronal differentiation in the cerebellum. Consistent with a role in adult neurogenesis, we showed expression of all tested receptor mRNAs in the proliferation zones around telencephalic ventricles at 2 mpf. This supports previous findings that suggested an important role of BMP and Activin A signaling in the maintenance and fate of NSC pools [35, 36].

In addition, we revealed strong and broad expression of all tested receptor mRNAs in various layers of the adult retina, which includes fully differentiated neurons and photoreceptors. Similar results were earlier obtained for rnAlk1, rnAcvr1 and rnBmpr2 in adult rat retinas [39]. In that study, it was proposed that these receptors contribute to the maintenance of tissue homeostasis, which could occur similarly in medaka. In the postnatal chick retina, it was shown that Smad1/5/8 signaling promotes formation of Müller glia progenitor cells (MGPCs) upon induced damage, whereas SMAD2/3 signaling suppresses MGPC formation [40]. However, other than for MGPCs, little is known about the function of BMPR signaling in photoreceptors and neurons of the retina. Their abundant expression in these cell types warrants future investigations to elucidate the underlying roles in retina tissue homeostasis.

## BMPRs in soft tissues and the skeletal system

The most prominent clinical symptoms of FOP are inflammatory soft tissue swellings, which turn soft tissues such as skeletal muscles or tendons into ectopic skeletal bone, which ultimately results in heterotopic ossifications (HOs) [41]. To recapitulate this phenotype in an animal model, the causative receptor complex consisting of hsACVR1-R206H and hsACVR2B has to be expressed in the respective tissue. Our study revealed the expression of *olaAlk1*, *olaAcvr1*, *olaAcvr1l* and *olaAcvr2bb* in myoseptal cells and within interstitial cells between myofibers. Interestingly, in the human and mouse muscle interstitium, a progenitor cell population of so-called fibro/adipogenic progenitors (FAPs) were described as potential cells of origin for HO [42]. Additionally, tendon-derived progenitors (Scx[+]) that mediate HO of ligaments and muscle resident interstitial progenitors (Mx1[+]) cells were identified to contribute to the diverse formations of extra skeletal bone [43]. Our observations that *olaAcvr1*, *olaAcvr1l* and *olaAcvr2bb* are expressed in myosepta and the muscle interstitium therefore highlight the potential of recapitulating HO in muscle and tendons in a FOP mutant medaka fish. Moreover, we observed *olaAcvr1*, *olaAcvr1l*, *olaAcvr2ba* and *olaBmpr2a* expression within the endothelial pillar cells of the gills. Interestingly, hsALK1, hsACVR1, hsBMPR2 and hsACVR2B are similarly expressed in the human vasculature [44]. Endothelial cells carrying the disease-causing receptor complex were associated with the onset of endothelial to mesenchymal transition (EndoMT), which upon inflammatory stimulus or changes in tissue environment can differentiate into chondrocytes and osteoblasts to induce HO [12, 45, 46]. In medaka, pillar cells within the gills express *olaAcvr1* and *olaAcvr1* as well as *olaAcvr2ba*, and

thus, when mutated, could possibly recapitulate aspects of FOP with formation of heterotopic cartilage and bone from soft tissue.

Whole mount *in situ* hybridization of 5 dpf embryos showed mRNA expression for all receptors in pectoral fins, which was limited to the mesenchyme but excluded from the apical ectodermal ridge (AER). In vertebrates, the AER controls limb outgrowth by stimulating proliferation of the underlying mesenchyme through paracrine secretion of several FGF, BMP (BMP2, BMP4 and BMP7) and WNT ligands [47, 48]. BMP signaling in the mesenchyme is required for initiation of chondrogenesis and formation of digit ray primordia in Sox9 positive progenitors [49]. Our findings in medaka match with previous mouse and chick studies that indicated BMPR expression in the fin mesenchyme. Interestingly, Bmpr1a is commonly described as the mediator of SMAD1/5 signaling in this context [50]. *olaAcvr1* receptor mRNAs are expressed in the pectoral fin mesenchyme, while Bmp2, Bmp4 and Bmp7 are secreted from the AER. Therefore, it is tempting to speculate that it is not only the Bmp2/4 high affinity receptor Bmpr1a that is involved in a SMAD 1/5 signaling complex, but also the Bmp6/7 high affinity receptor olaAcvr1 in combination with the type II receptors olaAcvr2a/b and olaBmpr2a.

Our analyses also highlighted differences in the expression of type I receptor genes *olaAcvr1* and *olaAcvr1l*, as well as for the type II receptor genes *olaAcvr2ab*, *olaAcvr2ba* and *olaAcvr2bb*. For example, 5 dpf embryos showed a more confined pattern of *olaAcvr1* expression when compared to *olaAcvr1l* in similar regions. In the caudal trunk region of 5 dpf embryos, *olaAcvr2ab* was expressed in both dorsal and ventral domains while *olaAcvr2ba* and *olaAcvr2bb* showed mutually exclusive expression in either dorsal or ventral regions. These different expression patterns suggest a distinct mode of spatiotemporal fine tuning of signaling for the respective paralogs.

## Evaluation of medaka Bmp/Activin receptor properties for a FOP model

Our sequence comparison revealed highly conserved Activin A binding interfaces between human hsACVR2B and the respective medaka orthologs. The conserved Activin A/ hsACVR2B binding interface [25] was functionally validated by a ligand binding assay, which exhibited similar ligand binding properties of hsACVR2 and the medaka orthologs. The Activin A high affinity receptor hsACVR2B is a key determinant in FOP triggering a hyperactive signaling complex by phosphorylating hsACVR1-R206H, which results in increased downstream signaling [51, 52]. By this, the gain of function mutation allows Activin A induced SMAD1/5/8 signaling [14]. Since hsACVR1 alone is a weak Activin A binder [24], hsACVR2B is crucial in mediating Activin A binding towards hsACVR1. Here, hsACVR2B binds hsActivin A with high affinity and subsequently interacts with T203 within the GS-box of hsACVR1 [52]. Our sequence comparison of the hsACVR1 kinase domain with that of the medaka orthologs revealed a high sequence conservation of the GS-box. Furthermore, R206 within the GS-box, which is mutated to histidine in FOP, is conserved in both medaka *olaAcvr1* and *olaAcvr1l*. Interestingly, a sequence comparison of the hsACVR1 LBD domain with that of olaAcvr1l and olaAcvr1 showed an overall lower sequence conservation when compared to the type II receptors. Additionally, sequence comparison and Western blot analysis revealed alternatively spliced transcript variants for both medaka *olaAcvr1* and *olaAcvr1l* encoding altered ligand binding domains. As a consequence, olaAcvr1l carries a prolonged N-terminus, while olaAcvr1 is carrying three instead of one LBD. While other transmembrane receptor families like VEGF receptors, Netrin receptors or LDL receptor-related proteins are known to exhibit repetitive extracellular domains that facilitate ligand binding [53–55], this so far has not been reported for BMP/Activin receptors. Hence, this could allow a different level of fine tuning or regulation of downstream signaling.

We also functionally characterized the two medaka olaAcvr1 receptors in a BRE reporter gene assay. This revealed the highest baseline BRE activity for olaAcvr1, for which BMP6 stimulation also showed the strongest effect. This could possibly be due to the fact that the three binding domains identified in olaAcvr1 exhibit a higher binding efficiency than a single domain. In contrast, transient expression of olaAcvr1l recapitulated the same baseline responsiveness as hsACVR1. When a R206H-equivalent mutation was introduced into both medaka olaAcvr1 variants, this recapitulated elevated luciferase activity when compared to wild-type receptors. Surprisingly, the respective R206H-like mutation in olaAcvr1 showed similar behavior as that in hsACVR1 despite different ligand binding domains. On the other hand, the olaAcvr1l receptor showed a significantly lower baseline luciferase activity and a stronger response to ligand stimulation. These differences could be due to the slightly less conserved GS-box and kinase domain of olaAcvr1l when compared to olaAcvr1.

In conclusion, in this study we showed co-expression of FOP relevant BMP type I and type II receptors in medaka tissues that are known to be affected in FOP-patients, such as endothelial tissues and myoseptal as well as muscle interstitial cells. Furthermore, we showed that Activin A ligand binding is conserved in medaka Acvr2 receptors and that the GS-box destabilizing R206H-equivalent mutations R379H in medaka olaAcvr1, and R239H in olaAcvr1l, respectively, resulted in elevated SMAD1/5/8 signaling, which is a hallmark of FOP. Importantly, since the medaka genome carries two *olaAcvr1* orthologs, introduction of a R206H-like mutation into only one of the two could prevent perinatal and embryonic lethality as earlier observed in mouse and zebrafish models, as the respective non-mutated ortholog could compensate lethality. As medaka has been shown to be a valuable model to study bone diseases, it will be worth generating a medaka FOP model. Whether the insertion of R206H-equivalent mutations into *olaAcvr1* or *olaAcvr1l* can recapitulate classical FOP phenotypes remains to be investigated.

## Materials and methods

### Fish lines and maintenance

All medaka experiments were conducted according to protocols approved by the Institutional Animal Care and Use Committee (IACUC) of the National University of Singapore (NUS; protocol numbers: R18-0562, R22-0470, BR19-0120). Fish were euthanized with an overdose of tricaine methanesulfonate (MS222; Merck) according to IACUC-approved protocols (SOP #303).

### Generation of *olaAcvr* riboprobes for RNA *in situ* hybridization

For riboprobe cloning, cDNA sequences for medaka Bmp receptor genes were retrieved from ENSEMBL databases (ensembl.org) with the following accession numbers: *olaAlk1*, ENSORLG00000007548; *olaAcvr1*, ENSORLG00000016722; *olaAcvr1l*, ENSORLG00000013834; *olaAcvr2ab*, ENSORLG00000000099; *olaAcvr2ba*, ENSORLG00000003784; *olaAcvr2bb*, ENSORLG00000019706; *olaBmpr2a* ENSORLG00000003034.

Riboprobes were generated as previously described (Tan and Winkler 2022) with slight modifications. For RNA extraction, whole medaka larvae (10 to 15 larvae; 18 dpf) or caudal fins (3 fins, at 2 mpf) were used. For generation of riboprobes, all cloning PCRs were carried out using Phusion Polymerase (Thermo Scientific F-530S) according to the manufacturer's guidelines (used primers are listed in S1 Table. PCR products were resolved by agarose gel electrophoresis and purified using a Promega Wizard SV and PCR Clean-up kit (A9282), according to the manufacturer's instructions. Next, a TOPO™ Cloning reaction (Invitrogen) was carried out according to the manufacturer's instructions. Plasmid DNA purification was

done according to the manufacturer's instructions (Promega-A1460). Sanger sequencing was conducted to validate the generated plasmids pTOPO-Alk1, pTOPO-Acvr1, pTOPO-Acvr1l, pTOPO-Acvr2ab, pTOPO- Acvr2ba, pTOPO- Acvr2bb and pTOPO-Bmpr2.

For riboprobe preparation, DNA plasmids were first linearized with *KpnI* or *NotI* (New England Biolabs) by overnight digestion at 37˚C. This was followed by *in vitro* transcription using T7 or SP6 RNA polymerases with 10x Digoxigenin (DIG) labeling mix (Roche), to generate sense and anti-sense riboprobes. After *in vitro* transcription, reaction mixtures were incubated with 2 units of Turbo DNase (Thermo Scientific) for 30 min at 37˚C followed by addition of 30 µl of 3.25 M LiCl (Thermo Scientific). Mixtures were incubated at -20˚C overnight and the precipitated RNA pelleted via centrifugation, dissolved in 25 µl of milliQ water and added to 75 µl of Hybridization mixture (HybMix) (50% formamide, 5x saline sodium citrate (SSC), 150 µg/ml heparin, 5 mg/ml torula RNA, 0.1%Tween 20) for storage at -20˚C as stock solutions. For all RNA *in situ* hybridization experiments, stock solutions were diluted at 1:50 with HybMix.

## Whole mount RNA *in situ* hybridization

Whole mount RNA *in situ* hybridizations for *olaAlk1*, *olaAcvr1*, *olaAcvr1l*, *olaAcvr2ab*, *olaAcvr2ba*, *olaAcvr2bb* and *olaBmpr2a* were performed on 5 dpf medaka embryos. For this, embryos were fixed in 4% paraformaldehyde (PFA; Sigma-Aldrich)/1x phosphate buffered saline (1x PBS pH 7.3; Sigma-Aldrich) overnight at 4˚C. After fixation, embryos were washed with 1x PBST (1x PBS with 0.1% v/v Tween 20) thrice (5 mins per wash), dechorionated manually using forceps and dehydrated in 100% MeOH for 5 mins at RT. Dehydrated embryos were transferred to a fresh solution of 100% MeOH and kept at -20˚C for at least 12 hours and up to two months. Rehydration of embryos was subsequently performed by incubation in a series of decreasing MeOH concentrations (75%, 50%, 25% and 0% in 1x PBST, 5 mins per solution) at RT on a shaker. Next, rehydrated embryos were treated with proteinase K (10 µg/ml; Thermo Scientific, diluted in 1x PBST) for 30 mins at RT. Proteinase K digestion was stopped by two washes with glycine (2 mg/ml in 1x PBST) and embryos were re-fixed in 4% PFA/1x PBS for 30 mins at RT. Re-fixed embryos were then washed with 1x PBST (5 times, 5 mins per wash) and prehybridized in HybMix at 65˚C for one hour. Riboprobes were denatured at 75˚C for 10 mins, chilled on ice and added to pre-hybridized embryos, which were then incubated at 65˚C overnight. Embryos were washed at 65˚C using a series of SSCT solutions (sequentially, SSCT I: 50% formamide, 2x SSC, 0.1% Tween 20; SSCT II: 2x SSC, 0.1% Tween 2; SSCT III: 0.2x SSC, 0.1% Tween 20; twice for each solution with 30 mins per wash) followed by a 5 mins 1x PBST wash at RT. Next, embryos were blocked in a 5% sheep serum/1x PBST solution for an hour at RT and incubated in the corresponding alkaline phosphatase coupled anti-DIG or anti-FLU antibody (Roche; 1:2000, diluted in blocking solution) at 4˚C overnight. Subsequently, embryos were washed with 1x PBST six times (20 mins per wash at RT on a rotator) and incubated in prestaining buffer (0.1 M NaCl, 0.05 M MgCl$_2$, 0.1 M Tris-HCl pH 9.5, 0.1% Tween 20) twice, 5 mins each incubation at RT. Embryos were then stained with an NBT/BCIP staining solution (Roche; 2% NBT/BCIP solution in 0.1 M NaCl, 0.1 M Tris-HCl pH 9.5, 0.1%Tween 20) in the dark at RT. Once staining was developed, embryos were washed with 1x PBST thrice (30 mins per wash) and mounted in 100% glycerol for imaging. Images were acquired using an Eclipse 90i, DIH Nikon microscope.

## Cryosectioning

Adult medaka tissues (2 mpf) were cryosectioned for subsequent RNA *in situ* hybridization. After euthanasia, fish tissues were dissected and fixed overnight in 4% PFA/ 1xPBS at 4˚C,

followed by three consecutive washes with 1x PBST (5 minutes per wash) at RT. Samples were then dehydrated in 100% MeOH for at least 12 hours and subsequently rehydrated in a series of decreasing MeOH concentrations (75%, 50%, 25% and 0% in 1x PBST; 5 minutes incubation per solution). Rehydrated tissues were embedded in 1.5% low melting agarose/ 5% sucrose/ 1xPBS on a petri dish. Agar blocks containing the samples were trimmed to the desired size and stored in 30% sucrose/1xPBS at 4˚C overnight. For sectioning of samples, a Leica CM1850 cryostat was pre-cooled to -25˚C. Specimen discs were pre-chilled in the cryostat and covered with a layer of frozen section medium (Leica, FSC 22). Embedded samples were placed on specimen discs with the frozen section medium and covered entirely with frozen medium and stored in the cooled cryostat for at least two hours. All samples were sectioned at 20 μm and collected on Superfrost® Plus microscope slides (Thermo Scientific). Sections were dried for at least 30 minutes at RT before storage at -20˚C.

## RNA *in situ* hybridization on cryosections

For RNA *in situ* hybridization on 20 μm cryosections, glass slides stored at -20˚C were thawed at 37˚C for an hour and rehydrated with 1x PBST for 15 minutes in a glass coplin jar at RT. After two subsequent washes with 1x PBS (5 mins per wash), samples were digested in a solution containing 5 μg/ml proteinase K (Thermo Scientific), 0.1 M Tris-HCl (pH 8.0) and 0.05 M EDTA. Next, sections were re-fixed in 4% PFA/1x PBS for 20 mins at RT. The refixed sections were then washed with 1x PBST (3 times, 5 minutes per wash) and acetylation was performed by incubation of sections for 5 mins in a freshly prepared solution containing 1.25% triethanolamine and 0.27% acetic anhydride diluted with milliQ water. Two consecutive washes with milliQ water (5 minutes per wash) were followed by incubation with 200 μl of the respective riboprobes (denatured at 75˚C for 10 minutes and chilled on ice) per glass slide. For incubation, slides were kept in a humidified sealed box in a 60˚C oven overnight. Next day, sections were washed twice for 30 minutes at 60˚C with a solution containing 50% formamide, 1x SSC and 0.1% Tween 20. Consecutively, sections were washed twice with 1 x PBST for 30 minutes and blocked for an hour in a 5% sheep serum/ 1x PBST solution and then incubated with an alkaline phosphatase coupled anti-DIG antibody (1:2000; diluted in blocking solution; Roche) in a humidified sealed box at 4˚C overnight. To remove unbound antibody, sections were washed in 1x TBST (0.14 M NaCl, 0.27 mM KCl, 25 mM Tris-HCl pH 7.3, 0.1% Tween 20) five times (20 minutes per wash at RT). Washed sections were then incubated two times with prestaining buffer as described above for 10 minutes at RT and stained with an NBT/ BCIP staining solution in the dark at RT or overnight at 4˚C till staining was developed. Stained sections were finally washed with 1x PBST thrice (10 minutes per wash) at RT and counterstained with DAPI (0.25 μg/ml; incubation for 10 minutes at RT) and mounted in Mowiol 4–88 (Calbiochem) for imaging. For better visibility, fluorescent DAPI was converted into blue signal on white background. Images were acquired using an Eclipse 90i, DIH Nikon microscope.

## Cloning for ligand surface binding assay

To generate a N-terminally Halo-tagged medaka BMP receptor library, previously generated pcDNA3.1:hsALK1-Halo and pcDNA3.1:hsACVR2B-Halo plasmids [24] were used as templates to insert *olaAlk1*, *olaAcvr1*, *olaAcvr1l*, as well as *olaAcvr2ab*, *olaAcvr2ba*, *olaAcvr2bb* and *olaBmpr2a*, respectively, in between *EcoRI* and *NotI* sites to replace the original receptor ORFs. Cloning PCRs were carried out on a Peltier Thermal Cycler (PTC-200) with primers listed in S2 Table. The elongation time was adapted according to the product size (Phusion Pol. = 1kb/min). PCR products were resolved by agarose gel electrophoresis, purified using a

NucleoSpin Gel and PCR Clean-up kit, restriction digested and ligated. Plasmids were validated by sequencing.

## Cell culture

COS-7 and HEK293T cells were obtained from the German Collection of Microorganisms and Cell Cultures (DSMZ) and cultivated in Dulbecco's Modified Eagle's Medium (DMEM) supplemented with 10% Fetal Calf Serum (FCS), 2 mM L-glutamine and penicillin (100 units/ml) / streptomycin (100 μg/ml) in a humidified atmosphere at 37°C and 5% CO2 (v/v). COS-7 cells were maintained in T175 flasks and cells were split 1:5 or 1:10, depending on given confluence.

## SDS-PAGE and Western-blotting

For sodium dodecyl sulfate polyacrylamide gel electrophoresis (SDS-PAGE), treated cells were lysed in 144 μl Laemmli buffer and frozen at -20°C. To ensure a homogeneous loading, cell lysates were pulled through a 1 ml syringe and boiled for 10 min at 95°C before loading onto 10% polyacrylamide gels. After gel-electrophoresis, proteins were transferred onto Methanol-activated PVDF membranes. Next, membranes were blocked for 1 hour in a solution containing 0.1% TBS-T and 3% w/v bovine serum albumin (BSA), then washed three times in 0.1% TBS-T and incubated with indicated primary antibodies overnight at 4°C. Primary antibodies: anti-GAPDH (Cell Signaling; #2118; monoclonal rabbit antibody), anti-Halo (ProMega; #G9211; monoclonal mouse antibody), were used at a 1:1,000 dilution in 3% w/v BSA/ TBS-T solution. For Horseradish peroxidase (HRP)-based detection, membranes were incubated with secondary goat-α-mouse or goat-α-rabbit IgG HRP conjugates (± 0.8 mg/ml, Dianova; #111-035-144, #115-035-068) for 1 hour at a dilution of 1:10,000. Chemiluminescent reactions were processed using WesternBright Quantum HRP substrate (Advansta) and documented on a FUSION FX7 digital imaging system.

## Dual luciferase reporter gene assay

HEK293T cells were transfected with Halo-tagged receptors together with a SMAD1/5/8 sensitive BRE2 luciferase reporter and ROR2 to analyze endogenous pSMAD1/5/8 levels. A construct constitutively expressing renilla luciferase (pGL4.74[hRluc/TK]; Promega) was co-transfected as internal control. The next day, cells were starved in serum-free medium for 3 h before stimulation with hBMP6 (2 nM) or hActivin A (5nM) for 24 h. Cell lysis was performed using passive lysis buffer (Promega) and measurement of luciferase activity was carried out according to manufacturer's instructions using a TECAN initiate f200 Luminometer (TECAN).

## Fluorescent growth factor labeling and ligand surface binding assay (LSBA)

Fluorescent growth factor labeling was performed as previously described [24]. For visualization of ActivinA-Cy5 ligand binding, 200.000 COS-7 cells/12 well were seeded in 1 ml DMEM full medium on glass cover slips (confocal: 18 mm or STED: precision #1.5H 10 mm diameter). On the following day, cells were transfected with desired constructs (500 ng DNA per well) using Lipofectamine 2000 according to the manufacturer's instructions. 24 hours post transfection cells were washed once with PBS and simultaneously incubated with the fluorescent Halo-ligand CA-Alexa Fluor 488 (Promega, #G1002) and 2 nM ActivinA-Cy5 for 30 minutes at 4°C to prevent internalization. Subsequently, cells were washed once with ice-cold PBS

before fixation with 100% methanol for 5 minutes. To remove the methanol, cells were washed again with PBS and finally mounted with Fluoromount G.

## Confocal microscopy

Confocal data of fixed COS-7 cells were acquired with the Expert Line STED Microscope from Abberior. Confocal images of COS-7 cells expressing Halo-tagged receptors stained with CA-Alexa488 and fluorescent ligand ActivinA-Cy5 were acquired using 485 nm (20% laser power) and 640 nm excitation (20% laser power). Image analysis and quantification was performed as previously described in [24].

## Statistical analysis and graphical design

All statistical tests were performed using GraphPad Prism version 9.3 software and are listed in the figure legends. Normal distribution of data sets was tested with the Shapiro-Wilk normality test. In cases of failure to reject the null hypothesis, the ANOVA and Tukey's post hoc test were used to check for statistical significance under the normality assumption. For all experiments statistical significance was assigned, with an alpha-level of $p < 0.05$. Figures were assembled using Adobe® Photoshop (Adobe Systems, San José, USA).

## Supporting information

**S1 Table. Oligonucleotides used for cloning of *in situ* riboprobes.**
(DOCX)

**S2 Table. Oligonucleotides used for cloning of Halo-tag plasmids.**
(DOCX)

**S1 Fig. Confirmation of functionality and specificity of generated *olaAlk1*, *olaAcvr1*, *olaAcvr1l*, *olaAcvr2ab*, *olaAcvr2ba*, *olaAcvr2bb* and *olaBmpr2a* anti-sense riboprobes used for *in situ* hybridization.** (A-G) Comparison of anti-sense riboprobe staining for *olaAlk1*, *olaAcvr1*, *olaAcvr1l*, *olaAcvr2ab*, *olaAcvr2ba*, *olaAcvr2bb and olaBmpr2a* with that of respective sense riboprobes (A'-G') by whole mount RNA *in situ* hybridization of 5 dpf medaka embryos. Arrows mark stained structures. Scale bars = 100 μm.
(TIF)

**S2 Fig. Expression of *olaAlk1*, *olaAcvr1*, *olaAcvr2ab*, *olaAcvr2ba*, *olaAcvr2bb* and *olaBmpr2a* in medaka head and brain tissue.** (A, A') *olaAlk1* expression pattern by whole mount RNA *in situ* hybridization in 5 dpf medaka embryos. (B) Overview of *olaAlk1* expression pattern in 2 mpf medaka head by RNA *in situ* hybridization on cryosections. (C, C') Magnified views of box marked in B, showing (C) *olaAlk1* expression and (C') DAPI staining (pseudocolored). (D-H) Comparison of *olaAcvr1*, *olaAcvr2ab*, *olaAcvr2ba*, *olaAcvr2bb* and *olaBmpr2a* expression in 2 mpf medaka head sections by RNA *in situ* hybridization. (I) *olaBmpr2a* expression pattern in 2 mpf medaka brain by RNA *in situ* hybridization on cryosections. ba—branchial arch, bb–basibranchial, bh–basihyal, ch–ceratohyal, gf—gill filaments, pq–palatoquadrate. Scale bars = 100 μm.
(TIF)

**S3 Fig. Expression of *olaAlk1* and *olaBmpr2a* in the retina of adult medaka.** Comparison of *olaAlk1* and *olaBmpr2a* expression in 2 mpf medaka eyes by RNA *in situ* hybridization on cryosections. (A, B) Overviews of respective receptor mRNA staining. (A'-B') Zoom in on area marked by red boxes in (A, B). Scale bars (A-B) = 100 μm, (A'-B') = 10 μm.
(TIF)

**S4 Fig. Expression of *olaAcvr1l*, *olaAcvr2ab*, *olaAcvr2ba*, *olaAcvr2bb*, *olaAlk1* and *olaBmpr2a* in adult medaka gills.** (A-F) Overview of *olaAcvr1l*, *olaAcvr2ab*, *olaAcvr2ba*, *olaAcvr2bb*, *olaAlk1* and *olaBmpr2a* expression by RNA *in situ* hybridization on cryosections. (G-H) Zoom-in view of *olaAlk1* and *olaBmpr2a* expression in 2 mpf medaka gills. ba–branchial arch, bb–basibranchial, bh–basihyal, ch–ceratohyal, ea–efferent artery, gf–gill filaments, h–heart, L–lamellae. Scale bars (A-F) = 100 μm, (G, H) = 15 μm.
(TIF)

**S5 Fig. Expression of *olaAlk1*, *olaAcvr2ab*, *olaAcvr2ba*, *olaAcvr2bb* and *olaBmpr2a* in trunk of embryonic and adult.** (A—D) Comparison of *olaAlk1* and *olaBmpr2a* expression by whole mount RNA *in situ* hybridization in 5 dpf medaka trunk and pectoral fin tissue. (E-H) Comparison of *olaAlk1*, *olaAcvr2ab*, *olaAcvr2ba*, *olaAcvr2bb* and *olaBmpr2a* expression in cryosections of 2 mpf medaka trunks by RNA *in situ* hybridization. aer–apical ectodermal ridge, fmc–fin mesenchyme, Myo–myosepta, Sc–spinal cord, Vc–vertebral column. Scale bars = 100 μm.
(TIF)

**S6 Fig. Conservation of receptor-binding interfaces on human and medaka Activin A and negative controls for hsActivin A-Cy5 binding.** (A) Alignment of hsActivin A ligand binding domain sequence with that of respective medaka ligands; polar receptor:ligand interaction sites are in light magenta and hydrophobic interaction sites in teal (PDB 1S4Y). (B) For negative control, transiently transfected COS-7 cells expressing Halo-tagged hsTGFBR2 receptors were simultaneously incubated with Halo-tag substrate CA-Alexa488 (green) and hsActivin A-Cy5 (magenta) at 24 hours post transfection (n = 3 independent experiments). (B upper left) Representative confocal microscopy images of COS-7 cells transiently expressing respective receptors, as well as untransfected controls. (B bottom) hsActivin A-Cy5 surface binding represented as relative fluorescence intensity per area to untransfected unstimulated control. Data is shown as fold induction ± standard deviation. Significance was calculated using two-way ANOVA and Tukey's post-hoc test. ***p < 0.001 ***, p <0.0001 ≡ significance as indicated (n = 3). (B upper right) Linear regression and correlation analysis of ligand:receptor binding based on Cy5-fluorescence intensity and normalized receptor fluorescence (CA-Alexa488) per area (n = 3). Correlation was analyzed using the Pearson Test (p < 0.0001 ≡ ****, p < 0.001 ≡ ***, p < 0.01 ≡ **, p < 0.5 ≡ *, p > 0.5 ≡ ns.). Scale bar = 20 μm.
(TIF)

**S7 Fig. Sequence alignment of human and medaka ACVR1 receptors and BRE-luciferase induction after receptor co-expression.** (A) Alignment of hsACVR1 ligand binding domain sequence with that of respective medaka receptors; similarities are highlighted in magenta. (B) Alignment of entire hsACVR1 kinase domain with that of respective medaka receptors; similarities are highlighted in magenta. (C) Luciferase activity one day after transfection of a SMAD1/5/8-sensitive BRE2-luciferase reporter and hRluc/TK together with either individual Halo-tagged hsACVR1, hsACVR1-R206H, olaAcvr1, olaAcvr1-R378H, olaAcvr1l, olaAcvr1l-R239H, or after co-transfection with hsACVR2B or olaAcvr2ba, respectively. ROR2 was used as control for endogenous signaling. After starvation for 3 h, HEK293T cells were stimulated overnight with hsActivin A (5 nM). Relative Luminescence Units (RLU) are expressed as mean fold induction ± standard deviation (n = 3 independent experiments). Statistical significance relative to no-stimulation control (w/o) was calculated using two-way ANOVA and Šidák multiple comparison test post-hoc test.
(TIF)

**S1 Raw images.**
(TIF)

## Acknowledgments

The authors would like to thank the Light Microscopy Unit at the NUS Centre for Bioimaging Sciences, and the fish facility staff at NUS Department of Biological Sciences for continued support.

## Author Contributions

**Conceptualization:** Michael Trumpp, Wen Hui Tan, Wiktor Burdzinski, Jerome Jatzlau, Petra Knaus, Christoph Winkler.

**Data curation:** Michael Trumpp, Wen Hui Tan, Wiktor Burdzinski, Yara Basler.

**Formal analysis:** Michael Trumpp, Wen Hui Tan, Wiktor Burdzinski, Yara Basler, Jerome Jatzlau, Petra Knaus, Christoph Winkler.

**Funding acquisition:** Petra Knaus, Christoph Winkler.

**Methodology:** Michael Trumpp, Wen Hui Tan, Wiktor Burdzinski, Yara Basler, Jerome Jatzlau, Christoph Winkler.

**Project administration:** Petra Knaus, Christoph Winkler.

**Resources:** Christoph Winkler.

**Supervision:** Wen Hui Tan, Jerome Jatzlau, Petra Knaus, Christoph Winkler.

**Validation:** Michael Trumpp, Wiktor Burdzinski, Jerome Jatzlau.

**Visualization:** Michael Trumpp, Wen Hui Tan, Wiktor Burdzinski, Jerome Jatzlau, Christoph Winkler.

**Writing – original draft:** Michael Trumpp, Jerome Jatzlau, Christoph Winkler.

**Writing – review & editing:** Wen Hui Tan, Wiktor Burdzinski, Yara Basler, Petra Knaus.

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
