## [Decision Letter · Decision Letter 0]

27 Mar 2023

PONE-D-22-34209Characterization of Acvr1/Acvr2 Activin receptors in medaka (Oryzias latipes): Towards establishing a novel animal model for Fibrodysplasia Ossificans ProgessivaPLOS ONE

Dear Dr. Winkler,

Thank you for submitting your manuscript to PLOS ONE. After careful consideration, we feel that it has merit but does not fully meet PLOS ONE’s publication criteria as it currently stands. Therefore, we invite you to submit a revised version of the manuscript that addresses the points raised during the review process.

We look forward to receiving your revised manuscript.

Kind regards,

Paul Eckhard Witten, PhD

Academic Editor

PLOS ONE

Journal Requirements:

“This work was funded by a NUS/BER Strategic Partnership grant (A-0004751-00-00) awarded from the Freie Universitaet Berlin and National University of Singapore to PK and CW, respectively. This work was further supported by the German Research Foundation DFG (SFB 1444) and by IPSEN GmbH to PK. CW is also supported by a grant from the Singapore Ministry of Education (MOE-T2EP30221-0008).”

“PK was funded by a NUS/BER Strategic Partnership grant (A-0004751-00-00) awarded from the Freie Universitaet Berlin.

PK was funded by German Research Foundation DFG (SFB 1444) and by IPSEN GmbH.

CW was funded by a NUS/BER Strategic Partnership grant (A-0004751-00-00) awarded from the National University of Singapore.

CW is funded by a grant from the Singapore Ministry of Education (MOE-T2EP30221-0008).

6. Please note that supplementary tables (should remain/ be uploaded) as separate "supporting information" files.

Additional Editor Comments:

Dear Dr. Winkler.

the reviewers provide positive and constructive comments on your manuscript. After minor revision it should be possible to accept the manuscript for publication.

Reviewer 1 has major concerns regarding the title of the manuscript and about considering g the animals already as a model for fibrodysplasia ossificans progressiva (FOP). At this stage, an animal model has, indeed, not been developed. Only after in-depth characterization of the phenotype and only if the phenotype reassembles the human disease condition a model would exist. Adult mutants are available, perhaps the readers could be informed why a characterization of the phenotype has not (not jet) been carried out. Reviewer 1 suggests to revise the introduction and the discussion of the manuscript accordingly. I agree with these suggestions. Reviewer 2 provides further details comments that should be useful to improve the manuscript.

I am looking forward to receive a revised version of the manuscript.

Best regards, Eckhard Witten

Reviewers' comments:

Reviewer's Responses to Questions

**Comments to the Author**

1. Is the manuscript technically sound, and do the data support the conclusions?

Reviewer #1: Partly

Reviewer #2: Yes

2. Has the statistical analysis been performed appropriately and rigorously? 

Reviewer #1: Yes

Reviewer #2: Yes

3. Have the authors made all data underlying the findings in their manuscript fully available?

Reviewer #1: Yes

Reviewer #2: Yes

4. Is the manuscript presented in an intelligible fashion and written in standard English?

Reviewer #1: Yes

Reviewer #2: Yes

5. Review Comments to the Author

Reviewer #1: The manuscript provides an in-depth characterization of the two activin receptors in the small teleost model Oryzias latipes. The expression and the functional implication are clearly stated and the results support the conclusions. My major concern is about promoting an animal model for fibrodysplasia ossificans progressiva (FOP), which is not strictly the scope of this paper. First of all, I would suggest changing the title, because is misleading. The authors have not yet developed an animal model and no studies on medaka bone are reported in the paper. Thus, although it is clear that the study represents the ’background work’ for establishing an animal model, it should be clearly specified (already in the title) that the work is only about the characterization of specific genes in a model species. As mentioned in the last paragraph of the abstract, it is preliminary work necessary before generating an animal model. This should be kept in mind throughout the manuscript. Thus, the scope of the introduction and the discussion should be reframed accordingly. Please see the attachment for additional comments.

Reviewer #2: The study describes the analysis of embryonic and adult expression and compares it with the expression of BMP and Activin type II receptors in medaka. In addition, the authors also analyzed Alk1 as a structurally related control for Acvr1, olaAcvr1l, and Bmpr2a as an Activin A low-affinity type II receptor.

The manuscript is well-written, and structured, and provides new and valuable information that medaka is a valuable model to study the disease Fibrodysplasia Ossificans Progressiva.

General and specific comments on the manuscript, that the authors may consider, are listed below.

-The authors in this study have determined the expression of ACVR1 orthologs olaAcvr1 and olaAcvr1l and also the Activin type II receptors olaAcvr2ab, olaAcvr2ba, and olaAcvr2bb in distinct embryonic and adult medaka tissues by in situ hybridization. The study could be complemented by qPCR analysis for the expression profile of these mRNA in embryos and adult tissues.

-Figure 1: the labels are hard to see, mainly the white letters over the white background in the figure.

-In the legend “md – midline” but in the figure, it appears “ml”

-In the legend of Figure 6 the meaning for “*”, “**”, and “****” is missing.

-The reference of Fig. S7B is missing or appears as Fig. S7A in line 259.

- In the legend of Figure S7 is missing the statistical analysis performed for the HEK293t cells that were stimulated overnight with hsActivin A relative to no-stimulation control (w/o).

- the abbreviations should be defined when they first appear in the text (e.g. ACVR1 in the Abstract; FCS in the Material and Methods)

- Uniformize the use of “μg/mL” or “μg/ml”.

- Uniformize the use of “HEK293T cells” or “HEK cells” or “HEK-cells” or “HEK293t cells”

- The plasmid RL-TK (in the material and Methods) is the same as RLTK-LUC (Figure legends)?

6. PLOS authors have the option to publish the peer review history of their article (what does this mean?). If published, this will include your full peer review and any attached files.

Reviewer #1: No

Reviewer #2: No

---

## [Author Response · Author response to Decision Letter 0]

2 Jun 2023

RESPONSE TO REVIEWERS

We thank both reviewers for their constructive and encouraging comments on our manuscript. We have addressed all concerns raised by the reviewers and highlighted all text changes in yellow in the manuscript file. We also listed our point by point responses in detail below.

Comments from Reviewer 1:

The manuscript provides an in-depth characterization of the two activin receptors in the small teleost model Oryzias latipes. The expression and the functional implication are clearly stated and the results support the conclusions. My major concern is about promoting an animal model for fibrodysplasia ossificans progressive (FOP), which is not strictly the scope of this paper. 

Authors’ response: We thank the reviewer for the encouraging and thoughtful comments.

Reviewer Comment 1: First of all, I would suggest changing the title, because is misleading. The authors have not yet developed an animal model and no studies on medaka bone are reported in the paper. Thus, although it is clear that the study represents the ’background work’ for establishing an animal model, it should be clearly specified (already in the title) that the work is only about the characterization of specific genes in a model species. As mentioned in the last paragraph of the abstract, it is preliminary work necessary before generating an animal model. This should be kept in mind throughout the manuscript. Thus, the scope of the introduction and the discussion should be reframed accordingly.

Authors’ response: We agree with the reviewer and have changed the title accordingly to: "Characterization of Fibrodysplaysia Ossificans Progressiva relevant Acvr1/Acvr2 Activin receptors in medaka (Oryzias latipes)". We also rephrased the text to emphasize that our study provides the groundwork for establishing a medaka FOP model (see changes highlighted in yellow in the text). 

Reviewer Comment 2: Line 59: TGFβ ‘-‘ missing (TGF-β)

Authors’ response: Corrected.

Reviewer Comment 3: The relationship between type I Activin receptors and SMAD activation is unclear and sometimes controversial in the introduction. Example: Lines 72-73 seem in contrast with lines 67-68. Previously ACVR1was referred to activate SMAD1/5/8 (line 68), later as 'unable to transmit SMAD signals (73)'.

Authors’ response: We agree that this text section was unclear and have now rephrased the text to better describe ACVR1 containing signaling and decoy complexes, as follows:

“While BMPs signal through BMP type I receptors (e.g., ACVR1, ACVR1L) to activate SMAD1/5/8 responses (Derynck and Zhang 2003; Katagiri and Watabe 2016), Activins signal through Activin type I receptors (ACVR1B/C) to induce the SMAD2/3 branch (Nickel and Mueller 2019). Under physiological conditions, human Activin A signals through a complex comprising the high affinity type II receptor hsACVR2A/B and low affinity type I receptor hsACVR1B (Cárcamo et al. 1994). If Activin A, however, binds to a complex that contains hsACVR1, this results in a decoy complex that is unable to transmit SMAD1/5/8 signals (Olsen et al. 2015; Valer et al. 2019; Aykul et al. 2020). As hsACVR1B competes with hsACVR1 for binding to common activin type II receptors, this provides a regulatory mechanism to control Activin A signal responses (Szilágyi et al. 2022). Only when activating mutations are present in the intracellular domain of hsACVR1, the decoy function of hsACVR1 is lost and Activin A induces SMAD1/5/8 signaling (Hatsell et al. 2015; Hildebrandt et al. 2021). For example, the most prominent hsACVR1 mutation in FOP, a substitution of arginine to histidine (R206H) in the glycine serine rich GS domain of the hsACVR1 kinase induces hyperactive SMAD1/5/8 signaling downstream of BMPs and Activin A (Chaikuad et al. 2012; Hatsell et al. 2015; Hino et al. 2015).”

Reviewer Comment 4: The authors should consider changing the order of some words to make reading easier and more fluent. Example: - line 82: 'to study the mutated FOP causing ACVR1 receptor, an...' change to 'to study the ACVR1 receptor causing FOP...'. Check throughout the manuscript.

Authors’ response: To make the text more readable, we changed the text for this and other sentences according to the reviewers’ suggestion. Changes are highlighted.

Reviewer Comment 5: Avoid very long sentences (i.e., lines 208-213).

Authors’ response: We changed the text according to the reviewer’s suggestion and shortened long sentences. For example, in line 218ff: “Together, these results suggest that olaAcvr1 and olaAcvr1l, the medaka orthologs of FOP-causing hsACVR1, as well as olaAcvr2ab, olaAcvr2ba and olaAcvr2bb, the orthologs of hsACVR2A/B encoding the corresponding interacting type II receptor, are mostly co-expressed in embryonic and adult tissues (summarized in Table 1). Most notably, they are expressed in soft tissues that get predominantly affected by FOP in humans.” Changes are highlighted

Reviewer Comment 6: Line 93. Specify 'other skeletal phenotypes' in max 1 sentence.

In the introduction, especially when describing the previous zebrafish models, it is mentioned that important insights into the disease were recapitulated, but no skeletal phenotype is described. The focus is rather set on molecular pathways. Please add 1-2 lines on what could be mimicked and what not to highlight the need for a new animal model.

Authors’ response: We agree with the reviewer. We refocused the introduction with a broader description of recapitulated FOP phenotypes, including skeletal phenotypes, in mice and fish model systems. We now also highlight the requirements for a FOP model as follows:

“To avoid this, chimeric or conditional approaches were used, which in mice resulted in heterotopic ossification, hind limb digit malformations and joint fusions (Chakkalakal et al. 2012; Hatsell et al. 2015). In zebrafish, a heat-shock inducible acvr1l_Q204D model exhibited severe skeletal phenotypes similar to those reported in human FOP patients such as heterotopic ossification, spinal lordosis, vertebral fusions, and malformed pelvic fins (LaBonty, Pray, and Yelick 2017). However, due to the inducible nature of mutant ACVR1 overexpression in both models, the spontaneous and progressive features of FOP could only be partially mimicked. To fully recapitulate a human FOP phenotype, an ideal model requires (1) expression of an endogenous ACVR1 allele carrying the R206H mutation during development, (2) retaining the spontaneous nature of disease onset, and (3) exhibiting mild clinical features already at birth. “

Reviewer Comment 7: Line 123: the acronym olaAlk1 is never explained in the manuscript. It is only mentioned that it is used as a control.

Authors’ response: We apologize for this oversight. We now explain olaAlk1 and describe its use as control for olaACVR1 in more detail as follows:

“As controls, we analyzed olaAlk1 encoding the ortholog of BMP type I receptor Activin receptor like 1 (ALK1/ACVRL1), which is the structurally closest relative of ACVR1, as well as the Activin A low affinity type II receptor olaBmpr2a.”

Reviewer Comment 8: Because most of the paper focuses on the expression pattern of medaka activin receptors type I and II, the introduction should contain some background info on what is known about the expression of the orthologues, at least in zebrafish, possibly in mouse. Other sections can be shortened (i.e. the section on invertebrates/D. melanogaster).

Authors’ response: As suggested by the reviewer, we removed the sentence on invertebrates. For mice and zebrafish, Activin type I and type II receptors are expressed in a wide range of cell types. Publicly available single cell RNAseq atlases describe the expression of these receptors in adult and embryonic mice and zebrafish. We added corresponding information in the text, as follows:

“The FOP-causing ACVR1 receptor was analyzed in mouse and zebrafish (Danio rerio, dr) to generate FOP models that mimic disease phenotype and progression (LaBonty and Yelick 2018; Kaliya-Perumal, Carney, and Ingham 2020). Single cell RNA sequencing of embryonic and adult mice and zebrafish revealed expression of Acvr1, Bmpr2 and Acvr2a/b orthologs in chondrocytes, mesoderm, myocytes, endothelial cells, and osteocytes (Farnsworth, Saunders, and Miller 2020; Cao et al. 2019; Schaum et al. 2018; Lange et al. 2023).”

Reviewer Comment 9: Results, I: The expression of olaAlk1 is detected in brain, branchial arches, blood vessels, ... 'close to skeletal structures'. Never observed expressed by skeletal structures in the head. How do you explain that? Maybe due to the observation timepoints?

Authors’ response: Among the BMP signaling receptors, activin A receptor like type 1 (ACVRL1, aka ALK1) has a predominant, cell-type specific expression in arterial endothelial cells (Roman and Hinck 2017; Panchenko et al. 1996; Oh et al. 2000; Roelen, van Rooijen, and Mummery 1997). This was shown for human, mouse and zebrafish (Seki, Yun, and Oh 2003; Robert et al. 2020). Therefore, expression of this receptor in vascularized tissues of medaka was expected. 

Reviewer Comment 10: Results, II: no expression of olaAcvr1/2 was detected in the gills at 5 days post-fertilization? If so, specify. 

Authors’ response: We apologize for the oversight not mentioning expression of olaAcvr1/2 in gills at 5 dpf. In fact, we did observe expression of olaAcvr1, olaAcvr1l, olaAcvr2ab, olaAcv2ba and olaAlk1 in branchial arches of the gills, which is shown in Fig. 1. This information is now added to the Results section as indicated below:

“Expression of BMP receptors in medaka is not limited to neuronal tissue. In embryonic gills at 5 dpf, olaAlk1, olaAcvr1, olaAcvr1l, olaAcvr2ab and olaAcvr2ba showed expression in branchial arches (Fig. 1A-E’, Fig. 1SA, A’). In gills of adult fish (2 mpf), olaAlk1, olaAcvr1, olaAcvr1l, olaAcvr2ab, olaAcvr2ba, olaAcvr2bb and olaBmpr2a showed expression in branchial arches, basibranchials, basihyals and gill filaments (Fig. 3A-E, Fig. S4A-H).” 

Reviewer Comment 11: line 189: specify what fins 

Authors’ response: The type of fins (i.e. pectoral fins) is now mentioned.

Reviewer Comment 12: Concerning the experiments in cells, it would have been more consistent to use a medaka soft tissue cell line instead of a monkey line.

Authors’ response: The reviewer is correct that having a medaka soft tissue cell line for in vitro comparison of the different medaka BMP receptors would have been the ideal approach. However, to the best of our knowledge, such a cell line is not available. On the other hand, we recently reported that the performance of the Ligand Surface Binding Assay (LSBA) is independent of the cellular context (Jatzlau et al. 2023). We tested the influence of different cell lines e.g., U2-OS or HEK293T cells, demonstrating that LSBA can be performed in multiple cell types with different endogenous receptor expression levels, obtaining similar results. 

Reviewer Comment 13: Line 493 should be euthanasia, not anesthesia, I hope. 

Authors’ response: We apologize for the mistake that has now been corrected.

Reviewer Comment 14: For in situ hybridization on non-decalcified sections, shouldn't the samples be treated with levamisole to avoid detection of endogenous ALP activity (i.e. in bone)?

Authors’ response: In our previous work, we had tested several already established medaka skeletal markers (e.g. osx, col1, col2, runx2, osc, etc.), as well as bone markers that we tested and described in medaka for the first time (col10a1, sparc, mmp13b and others) by RNA in-situ hybridization of decalcified whole mounts or tissue sections. In these studies, we never or rarely observed unspecific staining that would be indicative for endogenous ALP activity (e.g. see Renn et al. 2013; Renn et al. 2006; Renn and Winkler 2014; Yu et al. 2017). In addition, we also used sense controls in previous (Phan et al. 2020; Liu et al. 2022) as well as the present study (Fig. S1). These sense controls also exhibited no or at most low background staining in skeletal tissues. We therefore concluded that addition of levamisole is not necessary in our medaka RNA in-situ protocol.

Comments from Reviewer 2:

The study describes the analysis of embryonic and adult expression and compares it with the expression of BMP and Activin type II receptors in medaka. In addition, the authors also analyzed Alk1 as a structurally related control for Acvr1, olaAcvr1l, and Bmpr2a as an Activin A low-affinity type II receptor.

The manuscript is well-written, and structured, and provides new and valuable information that medaka is a valuable model to study the disease Fibrodysplasia Ossificans Progressiva.

General and specific comments on the manuscript, that the authors may consider, are listed below.

Authors’ response: We thank the reviewer for the positive and encouraging comments.

Reviewer Comment 1: The authors in this study have determined the expression of ACVR1 orthologs olaAcvr1 and olaAcvr1l and also the Activin type II receptors olaAcvr2ab, olaAcvr2ba, and olaAcvr2bb in distinct embryonic and adult medaka tissues by in situ hybridization. The study could be complemented by qPCR analysis for the expression profile of these mRNA in embryos and adult tissues.

Authors’ response: While we agree that qPCR would be the necessary approach to compare expression levels of various receptors at distinct stages, this analysis would lose aspects of spatial resolution, which was the main aim of our present study. qPCR analysis would not allow to identify single cells expressing receptors. For this, single cell RNA seq is necessary, which is beyond the scope of the current study.

We are currently working on a follow up study to establish a transgenic medaka FOP model. Once this is successful, we will perform qPCR analysis to compare expression levels of transgenic receptors with that of endogenous activin receptors. 

Reviewer Comment 2 and following: 

 -Figure 1: the labels are hard to see, mainly the white letters over the white background in the figure.

-In the legend “md – midline” but in the figure, it appears “ml”

-In the legend of Figure 6 the meaning for “*”, “**”, and “****” is missing.

-The reference of Fig. S7B is missing or appears as Fig. S7A in line 259.

- In the legend of Figure S7 is missing the statistical analysis performed for the HEK293t cells that were stimulated overnight with hsActivin A relative to no-stimulation control (w/o).

- the abbreviations should be defined when they first appear in the text (e.g. ACVR1 in the Abstract; FCS in the Material and Methods)

- Uniformize the use of “μg/mL” or “μg/ml”.

- Uniformize the use of “HEK293T cells” or “HEK cells” or “HEK-cells” or “HEK293t cells”

- The plasmid RL-TK (in the material and Methods) is the same as RLTK-LUC (Figure legends)?

Authors’ response: Thanks for highlighting these mistakes. All changes have been made according to the reviewer’s suggestions. 

References

Aykul, S., R. A. Corpina, E. J. Goebel, C. J. Cunanan, A. Dimitriou, H. J. Kim, Q. Zhang, A. Rafique, R. Leidich, X. Wang, J. McClain, J. Jimenez, K. C. Nannuru, N. J. Rothman, J. B. Lees-Shepard, E. Martinez-Hackert, A. J. Murphy, T. B. Thompson, A. N. Economides, and V. Idone. 2020. 'Activin A forms a non-signaling complex with ACVR1 and type II Activin/BMP receptors via its finger 2 tip loop', Elife, 9.

Cao, Junyue, Malte Spielmann, Xiaojie Qiu, Xingfan Huang, Daniel M. Ibrahim, Andrew J. Hill, Fan Zhang, Stefan Mundlos, Lena Christiansen, Frank J. Steemers, Cole Trapnell, and Jay Shendure. 2019. 'The single-cell transcriptional landscape of mammalian organogenesis', Nature, 566: 496-502.

Cárcamo, J., F. M. Weis, F. Ventura, R. Wieser, J. L. Wrana, L. Attisano, and J. Massagué. 1994. 'Type I receptors specify growth-inhibitory and transcriptional responses to transforming growth factor beta and activin', Mol Cell Biol, 14: 3810-21.

Chakkalakal, S. A., D. Zhang, A. L. Culbert, M. R. Convente, R. J. Caron, A. C. Wright, A. D. Maidment, F. S. Kaplan, and E. M. Shore. 2012. 'An Acvr1 R206H knock-in mouse has fibrodysplasia ossificans progressiva', J Bone Miner Res, 27: 1746-56.

Derynck, R., and Y. E. Zhang. 2003. 'Smad-dependent and Smad-independent pathways in TGF-beta family signalling', Nature, 425: 577-84.

Farnsworth, Dylan R., Lauren M. Saunders, and Adam C. Miller. 2020. 'A single-cell transcriptome atlas for zebrafish development', Developmental Biology, 459: 100-08.

Hatsell, S. J., V. Idone, D. M. Wolken, L. Huang, H. J. Kim, L. Wang, X. Wen, K. C. Nannuru, J. Jimenez, L. Xie, N. Das, G. Makhoul, R. Chernomorsky, D. D'Ambrosio, R. A. Corpina, C. J. Schoenherr, K. Feeley, P. B. Yu, G. D. Yancopoulos, A. J. Murphy, and A. N. Economides. 2015. 'ACVR1R206H receptor mutation causes fibrodysplasia ossificans progressiva by imparting responsiveness to activin A', Sci Transl Med, 7: 303ra137.

Hildebrandt, Susanne, Branka Kampfrath, Kristin Fischer, Laura Hildebrand, Julia Haupt, Harald Stachelscheid, and Petra Knaus. 2021. 'ActivinA Induced SMAD1/5 Signaling in an iPSC Derived EC Model of Fibrodysplasia Ossificans Progressiva (FOP) Can Be Rescued by the Drug Candidate Saracatinib', Stem Cell Reviews and Reports, 17: 1039-52.

Jatzlau, J., W. Burdzinski, M. Trumpp, L. Obendorf, K. Roßmann, K. Ravn, M. Hyvönen, F. Bottanelli, J. Broichhagen, and P. Knaus. 2023. 'A versatile Halo- and SNAP-tagged BMP/TGFβ receptor library for quantification of cell surface ligand binding', Commun Biol, 6: 34.

Kaliya-Perumal, A. K., T. J. Carney, and P. W. Ingham. 2020. 'Fibrodysplasia ossificans progressiva: current concepts from bench to bedside', Dis Model Mech, 13.

Katagiri, T., and T. Watabe. 2016. 'Bone Morphogenetic Proteins', Cold Spring Harb Perspect Biol, 8.

LaBonty, M., and P. C. Yelick. 2018. 'Animal models of fibrodysplasia ossificans progressiva', Dev Dyn, 247: 279-88.

Lange, Merlin, Alejandro Granados, Shruthi VijayKumar, Jordao Bragantini, Sarah Ancheta, Sreejith Santhosh, Michael Borja, Hirofumi Kobayashi, Erin McGeever, Ahmet Can Solak, Bin Yang, Xiang Zhao, Yang Liu, Angela Detweiler, Sheryl Paul, Honey Mekonen, Tiger Lao, Rachel Banks, Adrian Jacobo, Keir Balla, Kyle Awayan, Samuel D’Souza, Robert Haase, Alexandre Dizeux, Olivier Pourquie, Rafael Gómez-Sjöberg, Greg Huber, Mattia Serra, Norma Neff, Angela Oliveira Pisco, and Loïc A. Royer. 2023. 'Zebrahub – Multimodal Zebrafish Developmental Atlas Reveals the State Transition Dynamics of Late Vertebrate Pluripotent Axial Progenitors', bioRxiv: 2023.03.06.531398.

Liu, R., N. Imangali, L. P. Ethiraj, T. J. Carney, and C. Winkler. 2022. 'Transcriptome Profiling of Osteoblasts in a Medaka (Oryzias latipes) Osteoporosis Model Identifies Mmp13b as Crucial for Osteoclast Activation', Front Cell Dev Biol, 10: 775512.

Nickel, J., and T. D. Mueller. 2019. 'Specification of BMP Signaling', Cells, 8.

Oh, S. P., T. Seki, K. A. Goss, T. Imamura, Y. Yi, P. K. Donahoe, L. Li, K. Miyazono, P. ten Dijke, S. Kim, and E. Li. 2000. 'Activin receptor-like kinase 1 modulates transforming growth factor-beta 1 signaling in the regulation of angiogenesis', Proc Natl Acad Sci U S A, 97: 2626-31.

Olsen, O. E., K. F. Wader, H. Hella, A. K. Mylin, I. Turesson, I. Nesthus, A. Waage, A. Sundan, and T. Holien. 2015. 'Activin A inhibits BMP-signaling by binding ACVR2A and ACVR2B', Cell Communication and Signaling, 13.

Panchenko, M. P., M. C. Williams, J. S. Brody, and Q. Yu. 1996. 'Type I receptor serine-threonine kinase preferentially expressed in pulmonary blood vessels', Am J Physiol, 270: L547-58.

Phan, Q. T., R. Liu, W. H. Tan, N. Imangali, B. Cheong, M. Schartl, and C. Winkler. 2020. 'Macrophages Switch to an Osteo-Modulatory Profile Upon RANKL Induction in a Medaka (Oryzias latipes) Osteoporosis Model', JBMR Plus, 4: e10409.

Renn, J., A. Buttner, T. T. To, S. J. Chan, and C. Winkler. 2013. 'A col10a1:nlGFP transgenic line displays putative osteoblast precursors at the medaka notochordal sheath prior to mineralization', Dev Biol, 381: 134-43.

Renn, J., M. Schaedel, J. N. Volff, R. Goerlich, M. Schartl, and C. Winkler. 2006. 'Dynamic expression of sparc precedes formation of skeletal elements in the Medaka (Oryzias latipes)', Gene, 372: 208-18.

Renn, J., and C. Winkler. 2014. 'Osterix/Sp7 regulates biomineralization of otoliths and bone in medaka (Oryzias latipes)', Matrix Biol, 34: 193-204.

Robert, F., A. Desroches-Castan, S. Bailly, S. Dupuis-Girod, and J. J. Feige. 2020. 'Future treatments for hereditary hemorrhagic telangiectasia', Orphanet J Rare Dis, 15: 4.

Roelen, B. A., M. A. van Rooijen, and C. L. Mummery. 1997. 'Expression of ALK-1, a type 1 serine/threonine kinase receptor, coincides with sites of vasculogenesis and angiogenesis in early mouse development', Dev Dyn, 209: 418-30.

Roman, B. L., and A. P. Hinck. 2017. 'ALK1 signaling in development and disease: new paradigms', Cell Mol Life Sci, 74: 4539-60.

Schaum, Nicholas, Jim Karkanias, Norma F. Neff, Andrew P. May, Stephen R. Quake, Tony Wyss-Coray, Spyros Darmanis, Joshua Batson, Olga Botvinnik, Michelle B. Chen, Steven Chen, Foad Green, Robert C. Jones, Ashley Maynard, Lolita Penland, Angela Oliveira Pisco, Rene V. Sit, Geoffrey M. Stanley, James T. Webber, Fabio Zanini, Ankit S. Baghel, Isaac Bakerman, Ishita Bansal, Daniela Berdnik, Biter Bilen, Douglas Brownfield, Corey Cain, Michelle B. Chen, Steven Chen, Min Cho, Giana Cirolia, Stephanie D. Conley, Spyros Darmanis, Aaron Demers, Kubilay Demir, Antoine de Morree, Tessa Divita, Haley du Bois, Laughing Bear Torrez Dulgeroff, Hamid Ebadi, F. Hernán Espinoza, Matt Fish, Qiang Gan, Benson M. George, Astrid Gillich, Foad Green, Geraldine Genetiano, Xueying Gu, Gunsagar S. Gulati, Yan Hang, Shayan Hosseinzadeh, Albin Huang, Tal Iram, Taichi Isobe, Feather Ives, Robert C. Jones, Kevin S. Kao, Guruswamy Karnam, Aaron M. Kershner, Bernhard M. Kiss, William Kong, Maya E. Kumar, Jonathan Y. Lam, Davis P. Lee, Song E. Lee, Guang Li, Qingyun Li, Ling Liu, Annie Lo, Wan-Jin Lu, Anoop Manjunath, Andrew P. May, Kaia L. May, Oliver L. May, Ashley Maynard, Marina McKay, Ross J. Metzger, Marco Mignardi, Dullei Min, Ahmad N. Nabhan, Norma F. Neff, Katharine M. Ng, Joseph Noh, Rasika Patkar, Weng Chuan Peng, Lolita Penland, Robert Puccinelli, Eric J. Rulifson, Nicholas Schaum, Shaheen S. Sikandar, Rahul Sinha, Rene V. Sit, Krzysztof Szade, Weilun Tan, Cristina Tato, Krissie Tellez, Kyle J. Travaglini, Carolina Tropini, Lucas Waldburger, Linda J. van Weele, Michael N. Wosczyna, Jinyi Xiang, Soso Xue, Justin Youngyunpipatkul, Fabio Zanini, Macy E. Zardeneta, Fan Zhang, Lu Zhou, Ishita Bansal, Steven Chen, Min Cho, Giana Cirolia, Spyros Darmanis, Aaron Demers, Tessa Divita, Hamid Ebadi, Geraldine Genetiano, Foad Green, Shayan Hosseinzadeh, Feather Ives, Annie Lo, Andrew P. May, Ashley Maynard, Marina McKay, Norma F. Neff, Lolita Penland, Rene V. Sit, Weilun Tan, Lucas Waldburger, Justin Youngyunpipatkul, Joshua Batson, Olga Botvinnik, Paola Castro, Derek Croote, Spyros Darmanis, Joseph L. DeRisi, Jim Karkanias, Angela Oliveira Pisco, Geoffrey M. Stanley, James T. Webber, Fabio Zanini, Ankit S. Baghel, Isaac Bakerman, Joshua Batson, Biter Bilen, Olga Botvinnik, Douglas Brownfield, Michelle B. Chen, Spyros Darmanis, Kubilay Demir, Antoine de Morree, Hamid Ebadi, F. Hernán Espinoza, Matt Fish, Qiang Gan, Benson M. George, Astrid Gillich, Xueying Gu, Gunsagar S. Gulati, Yan Hang, Albin Huang, Tal Iram, Taichi Isobe, Guruswamy Karnam, Aaron M. Kershner, Bernhard M. Kiss, William Kong, Christin S. Kuo, Jonathan Y. Lam, Benoit Lehallier, Guang Li, Qingyun Li, Ling Liu, Wan-Jin Lu, Dullei Min, Ahmad N. Nabhan, Katharine M. Ng, Patricia K. Nguyen, Rasika Patkar, Weng Chuan Peng, Lolita Penland, Eric J. Rulifson, Nicholas Schaum, Shaheen S. Sikandar, Rahul Sinha, Krzysztof Szade, Serena Y. Tan, Krissie Tellez, Kyle J. Travaglini, Carolina Tropini, Linda J. van Weele, Bruce M. Wang, Michael N. Wosczyna, Jinyi Xiang, Hanadie Yousef, Lu Zhou, Joshua Batson, Olga Botvinnik, Steven Chen, Spyros Darmanis, Foad Green, Andrew P. May, Ashley Maynard, Angela Oliveira Pisco, Stephen R. Quake, Nicholas Schaum, Geoffrey M. Stanley, James T. Webber, Tony Wyss-Coray, Fabio Zanini, Philip A. Beachy, Charles K. F. Chan, Antoine de Morree, Benson M. George, Gunsagar S. Gulati, Yan Hang, Kerwyn Casey Huang, Tal Iram, Taichi Isobe, Aaron M. Kershner, Bernhard M. Kiss, William Kong, Guang Li, Qingyun Li, Ling Liu, Wan-Jin Lu, Ahmad N. Nabhan, Katharine M. Ng, Patricia K. Nguyen, Weng Chuan Peng, Eric J. Rulifson, Nicholas Schaum, Shaheen S. Sikandar, Rahul Sinha, Krzysztof Szade, Kyle J. Travaglini, Carolina Tropini, Bruce M. Wang, Kenneth Weinberg, Michael N. Wosczyna, Sean M. Wu, Hanadie Yousef, Ben A. Barres, Philip A. Beachy, Charles K. F. Chan, Michael F. Clarke, Spyros Darmanis, Kerwyn Casey Huang, Jim Karkanias, Seung K. Kim, Mark A. Krasnow, Maya E. Kumar, Christin S. Kuo, Andrew P. May, Ross J. Metzger, Norma F. Neff, Roel Nusse, Patricia K. Nguyen, Thomas A. Rando, Justin Sonnenburg, Bruce M. Wang, Kenneth Weinberg, Irving L. Weissman, Sean M. Wu, Stephen R. Quake, Tony Wyss-Coray, Consortium The Tabula Muris, coordination Overall, coordination Logistical, collection Organ, processing, preparation Library, sequencing, analysis Computational data, annotation Cell type, group Writing, group Supplemental text writing, and investigators Principal. 2018. 'Single-cell transcriptomics of 20 mouse organs creates a Tabula Muris', Nature, 562: 367-72.

Seki, T., J. Yun, and S. P. Oh. 2003. 'Arterial endothelium-specific activin receptor-like kinase 1 expression suggests its role in arterialization and vascular remodeling', Circ Res, 93: 682-9.

Szilágyi, S. S., A. R. Amsalem-Zafran, K. E. Shapira, M. Ehrlich, and Y. I. Henis. 2022. 'Competition between type I activin and BMP receptors for binding to ACVR2A regulates signaling to distinct Smad pathways', Bmc Biology, 20: 50.

Valer, J. A., C. Sánchez-de-Diego, C. Pimenta-Lopes, J. L. Rosa, and F. Ventura. 2019. 'ACVR1 Function in Health and Disease', Cells, 8.

Yu, T., M. Graf, J. Renn, M. Schartl, D. Larionova, A. Huysseune, P. E. Witten, and C. Winkler. 2017. 'A vertebrate-specific and essential role for osterix in osteogenesis revealed by gene knockout in the teleost medaka', Development, 144: 265-71.

---

## [Decision Letter · Decision Letter 1]

12 Jul 2023

PONE-D-22-34209R1Characterization of Fibrodysplasia Ossificans Progessiva relevant Acvr1/Acvr2 Activin receptors in medaka (Oryzias latipes)PLOS ONE

Dear Dr. Winkler,

Thank you for submitting your manuscript to PLOS ONE. On item is left that needs to be corrected. After this correction the  manuscript  should ready for  publication.

We received a positive comments on the revised version of the manuscript. There is one item left that needs to be corrected. Reviewer 1 (comment 14) comments on the visualization of the in situ hybridization. The reviewer asks why no levamisole has been used to block endogenous ALP activity. In deed without levamisole one would expect endogenous ALP activity. Perhaps the samples have been stored in formalin for longer time, a procedure that would inactivate endogenous ALP. If so please provide the details.

Concerning the details of the in situ hybridization visualization step:

- The material and methods sections refers to Renn & Winkler 2009 for details.

- Renn & Winkler 2009 refer to Renn et al. 2006 for details.

- Renn et al. 2006 refer to Klüver et al. 2005 for details.

- Klüver et al. 2005 do not provide details about the visualization step of the in situ protocol, end of story.

This can of course not be accepted. Best is to provide the reader directly with the protocol and if here is a reference to a second publication this publication must provide all required details without reference to a third publication. Please correct the in situ hybridization protocol. After the correction the manuscript should be ready for publication.

We look forward to receiving your revised manuscript.

Kind regards,

Paul Eckhard Witten, PhD

Academic Editor

PLOS ONE

Journal Requirements:

Reviewers' comments:

Reviewer's Responses to Questions

**Comments to the Author**

1. If the authors have adequately addressed your comments raised in a previous round of review and you feel that this manuscript is now acceptable for publication, you may indicate that here to bypass the “Comments to the Author” section, enter your conflict of interest statement in the “Confidential to Editor” section, and submit your "Accept" recommendation.

Reviewer #1: All comments have been addressed

2. Is the manuscript technically sound, and do the data support the conclusions?

Reviewer #1: Yes

3. Has the statistical analysis been performed appropriately and rigorously? 

Reviewer #1: Yes

4. Have the authors made all data underlying the findings in their manuscript fully available?

Reviewer #1: Yes

5. Is the manuscript presented in an intelligible fashion and written in standard English?

Reviewer #1: Yes

6. Review Comments to the Author

Reviewer #1: Thank you for having carefully addressed all of my concerns. Concerning in situ hybridization, I would suggest inhibiting endogenous ALP for future works, even if sense controls have been used.

7. PLOS authors have the option to publish the peer review history of their article (what does this mean?). If published, this will include your full peer review and any attached files.

Reviewer #1: No

---

## [Author Response · Author response to Decision Letter 1]

9 Aug 2023

As requested, we have included a detailed protocol for the used whole mount RNA in-situ hybridization (highlighted in yellow).

---

## [Editor Report · Decision Letter 2]

29 Aug 2023

Characterization of Fibrodysplasia Ossificans Progessiva relevant Acvr1/Acvr2 Activin receptors in medaka (Oryzias latipes)

PONE-D-22-34209R2

Dear Dr. Winkler,

We’re pleased to inform you that your manuscript has been judged scientifically suitable for publication and will be formally accepted for publication once it meets all outstanding technical requirements.

Kind regards,

Paul Eckhard Witten, PhD

Academic Editor

PLOS ONE
---

## [Editor Report · Acceptance letter]

5 Sep 2023

PONE-D-22-34209R2 

Characterization of Fibrodysplasia Ossificans Progessiva relevant Acvr1/Acvr2 Activin receptors in medaka (*Oryzias latipes*) 

Dear Dr. Winkler:

I'm pleased to inform you that your manuscript has been deemed suitable for publication in PLOS ONE. Congratulations! Your manuscript is now with our production department. 

Kind regards, 

on behalf of

Dr. Paul Eckhard Witten 

Academic Editor

PLOS ONE